# *LAST*, a c-Myc-inducible long noncoding RNA, cooperates with CNBP to promote *CCND1* mRNA stability in human cells

Limian Cao[1†], Pengfei Zhang[1†], Jinming Li[2], Mian Wu[1,2*]

[1]CAS Key Laboratory of Innate Immunity and Chronic Disease, CAS Center for Excellence in Molecular Cell Science, Innovation Center for Cell Signaling Network, School of Life Sciences, University of Science & Technology of China, Hefei, China; [2]Translational Research Institute, Henan Provincial People's Hospital, School of Medicine, Henan University, Zhengzhou, China

**Abstract** Cyclin D1 is a critical regulator of cell cycle progression and works at the G1 to S-phase transition. Here, we report the isolation and characterization of the novel c-Myc-regulated lncRNA *LAST* (LncRNA-Assisted Stabilization of Transcripts), which acts as a *CCND1* mRNA stabilizer. Mechanistically, *LAST* was shown to cooperate with CNBP to bind to the 5′UTR of *CCND1* mRNA to protect against possible nuclease targeting. In addition, data from CNBP RIP-seq and *LAST* RNA-seq showed that *CCND1* mRNA might not be the only target of *LAST* and CNBP; three additional mRNAs were shown to be post-transcriptional targets of *LAST* and CNBP. In a xenograft model, depletion of *LAST* diminished and ectopic expression of *LAST* induced tumor formation, which are suggestive of its oncogenic function. We thus report a previously unknown lncRNA involved in the fine-tuned regulation of *CCND1* mRNA stability, without which *CCND1* exhibits, at most, partial expression.
DOI: https://doi.org/10.7554/eLife.30433.001

**\*For correspondence:**
wumian@ustc.edu.cn

[†]These authors contributed equally to this work

**Competing interests:** The authors declare that no competing interests exist.

## Introduction

The oncoprotein c-Myc plays a pivotal role in multiple cellular processes, such as cell cycle progression, malignant transformation, differentiation suppression and apoptosis induction, predominantly through its transcription activity (*Seth et al., 1993*; *Drayton et al., 2003*; *Wei et al., 2003*; *Demeterco et al., 2002*; *Prendergast, 1999*; *Amati et al., 1992*; *Lee et al., 1996*; *Hoffman and Liebermann, 2008*). Indeed, as a master transcriptional factor, c-Myc regulates the expression of approximately 10–15% of genes in the genome, including a variety of protein-coding genes (*Lin et al., 2012*; *Nie et al., 2012*; *Fernandez et al., 2003*), such as *CDKN1A*, *CDKN2B*, *CCND1*, *CCND2*, *CDK4* and *E2F2* (*Adhikary and Eilers, 2005*).

Among c-Myc target genes, *CCND1* is of particular importance in cell cycle control and is characterized by the dramatic periodicity of the abundance of its protein product cyclin D1 throughout the cell cycle (*Sherr, 1995*). Cyclin D1 forms a complex with CDK4 or CDK6 and functions as a regulatory subunit whose activity is required for G1/S transition (*Sherr, 1995*; *Resnitzky et al., 1994*). Cyclin D1 also interacts with the tumor suppressor pRB1, which in turn positively regulates cyclin D1 expression (*DeGregori, 2004*). Mutation, amplification and overexpression of *CCND1* are frequently observed in cancer and have been reported to contribute to tumorigenesis (*Wiestner et al., 2007*; *Elsheikh et al., 2008*; *Musgrove et al., 2011*). Cyclin D1 is a short-lived protein with a rapid turnover rate (~24 min) due to degradation by the ubiquitin-proteasome system (*Diehl et al., 1998*; *Diehl et al., 1997*). While early studies showed that the Skp2 F-box protein is involved in cyclin D1 degradation (*Yu et al., 1998*), a recent study has identified two additional F-box proteins that play

**eLife digest** Cell division involves a series of steps in which the cell grows, duplicates its contents, and then divides into two. Together these steps are called the cell cycle, and the transition between each step must be controlled to make sure that events take place in the right order. Any loss of control can cause cells to divide in an unrestrained manner, which may lead to cancer.

Proteins called cyclins control progression through the cell cycle. As such, these proteins need to be produced in the correct amounts and at the correct times. Transcription factors are proteins that switch genes on or off to help regulate how much protein is made from those genes. A transcription factor known as c-Myc regulates the expression of the genes that encode the cyclins. Among these genes, one called CCND1 is particularly important because it encodes a protein that controls a crucial transition in the cell cycle: it marks a 'point of no return', beyond which cells are committed to dividing.

When a transcription factor switches on a gene, the gene gets copied into a molecule of messenger RNA, which is then translated into protein. But, cells also contain genes that do not code for proteins. Transcription factors can bind to such non-coding genes, leading to the production of so-called long non-coding RNAs (often abbreviated to lncRNAs).

Many lncRNAs can affect the expression of other genes. Cao, Zhang et al. have now asked whether any lncRNAs regulate CCND1 in human cells. The analysis revealed that the transcription factor c-Myc promotes the expression of a previously unidentified lncRNA. Cao, Zhang et al. name this lncRNA LAST, which is officially short for LncRNA-assisted stabilization of transcripts, and show thatit makes the CCND1 messenger RNA more stable. In other words, it makes the messenger RNAs 'last' longer in the cell. This in turn, ensures that the cell cycle progresses in the correct manner, allowing cells to complete their division. In the absence of LAST, the CCND1 messenger RNA becomes unstable and as a result the cell cycle does not progress.

Cao, Zhang et al. then explored the role of LAST in cancer cells. When human colon cancer cells that expressed LAST were implanted into mice, they formed tumors. Yet, reducing the expression of LAST in the colon cancer cells made the tumors grow slower.

Future challenges will be to understand how LAST makes messenger RNAs stable and further explore its role in cancer. A better understanding of this molecule could reveal whether it can be used to help doctors diagnose or treat cancers.

DOI: https://doi.org/10.7554/eLife.30433.002

important roles in targeting cyclin D1 for proteasome degradation (*Lin et al., 2006*; *Okabe et al., 2006*).

c-Myc can upregulate or downregulate expression of cyclin D1 in a context-dependent manner. On the one hand, c-Myc, together with Max, a co-transcription factor, activates CCND1 transcription through an E box located at −558 nt in its promoter (*Kress et al., 2015*; *Yu et al., 2005*; *Guo et al., 2011*). On the other hand, simultaneous overexpression of c-Myc and ZO-2 enhances repression of the CCND1 promoter through the E box in MDCK cells (*Gonzalez-Mariscal et al., 2009*). In addition, c-Myc has been reported to repress the cyclin DI promoter and antagonize USF-mediated transactivation in BALB/c-3T3, Rat6 and rat embryo fibroblasts (*Philipp et al., 1994*). In addition to c-Myc, multiple transcription factors, including AP-1, NF-κB, E2F and Oct-1, can bind to their respective CCND1 promoters and regulate its expression (*Guo et al., 2011*). CCND1 can also be regulated epigenetically through histone modifications; GATA3 cooperates with PARP1 to regulate CCND1 transcription by modulating histone H1 incorporation (*Shan et al., 2014*). Moreover, post-transcriptional mechanisms are also involved in the regulation of CCND1, as exemplified by MYF5-mediated enhancement in CCND1 mRNA translation, which contributes to early myogenesis (*Panda et al., 2016*). Mutations in CCND1 leading to stable truncated transcripts are associated with increased cell proliferation and shortened survival of cancer patients (*Wiestner et al., 2007*).

Long noncoding RNAs (lncRNAs), which are defined as transcripts that are longer than 200 nucleotides and lack protein coding capacity, are emerging as important regulators of biological processes, including regulation of gene expression at multiple levels, such as chromatin remodeling, transcription, and post-transcriptional modulation (*Derrien et al., 2012*; *Iorio and Croce, 2012*;

*Bonasio and Shiekhattar, 2014*; *Wilusz et al., 2008*). Genome-wide studies have shown that c-Myc transcriptionally regulates many lncRNA genes, such as *PVT1*, the CCAT family, and MYCLos, whereas a number of lncRNAs have been demonstrated to be important components of the c-Myc-mediated signaling network (*Colombo et al., 2015*; *Nissan et al., 2012*; *Ling et al., 2013*; *Kim et al., 2015a*; *Kim et al., 2015b*). Nevertheless, whether lncRNAs participate in the regulation of *CCND1* remains to be fully studied.

Here, we report the isolation and characterization of the novel c-Myc regulated *LAST*, which acts as a *CCND1* mRNA stabilizer and without which *CCND1* mRNA becomes unstable and cell cycle arrest occurs in the G1 phase. Mechanistically, *LAST* cooperates with CNBP, a single-stranded DNA/RNA-binding factor, to bind to the 5' untranslated region of *CCND1* messenger RNA, possibly to protect against nuclease degradation. This report describes a model by which lncRNA stabilizes mRNA post-transcriptionally via 5'-end protection.

## Results

### Identification of *LAST*, a c-Myc-responsive long noncoding RNA that promotes cell proliferation

To identify novel functions of c-Myc-regulated long non-coding RNAs, doxycycline-treated or untreated P493-6 cells carrying a c-Myc tet-off system (*Kim et al., 2007*) were used to analyze the lncRNA expression profile via long non-coding RNA microarray analysis (*Supplementary file 1*, GSE106916). We selected five significantly c-Myc-downregulated lncRNAs (fold change above 8, P-value below 0.01) that were identified by the lncRNA microarray. Two of the five lncRNAs, namely, lncRNA-51 and lncRNA-52, along with CDK4 (positive control) were found to be downregulated when c-Myc expression was suppressed by doxycycline treatment (*Figure 1A*). Of these two c-Myc responsive lncRNAs, lncRNA-52 (RP11-660L16.2, ENST00000529369) was chosen for further investigation because knockdown of this lncRNA (*Figure 1B*) showed a *significant reduction in colony formation* (*Figure 1C and D*). lncRNA-52 is located approximately 1.8 Mb downstream of the *cyclin D1/CCND1* gene in a head-to-tail orientation (*Figure 1—figure supplement 1A*). As will be shown in the following sections, this lncRNA is able to promote the stability of mRNA transcripts, including *CCND1* mRNA; we therefore named it *LAST* (LncRNA-Assisted Stabilization of Transcripts).

To verify the existence of endogenous*LAST* and to determine its molecular size, northern blot analysis was performed. A band of approximately 700 nt in length was detected in both P493-6 and H1299 cells, but was absent in P493-6 cells treated with doxycycline, which suppresses c-Myc expression (*Figure 1E*). The apparent size of *LAST* was the same as predicted by the UCSC (University of California, Santa Cruz) Genome Browser, suggesting that *LAST* is a full-length transcript. To investigate the cellular compartment in which *LAST* is located, single molecule RNA fluorescent in situ hybridization (FISH) was performed in both HCT116 and H1299 cells. As shown in *Figure 1F*, *LAST* was predominantly localized in the cytosol. Cytosol localization of *LAST* was also confirmed by determining the levels of *LAST* in different sub-cellular fractions (*Figure 1G*). Moreover, the signal intensity of *LAST* was reduced in *LAST*-depleted cells (*Figure 1—figure supplement 1B*). It has been reported that lncRNAs are often present at relatively low copy numbers; hence, we measured the *LAST* transcript copy number per cell in various cell lines, including the normal cell lines HAFF, IMR90, and MCF10A and tumor cell lines HCT116, MCF7 and H1299. The *LAST* copy number was higher in tumor cells than in normal cells (*Figure 1H*).

Lentivirus-mediated gene knockdown of c-Myc decreased whereas ectopic expression of c-Myc increased *LAST* expression in HCT116, H1299 and MCF10A cells (*Figure 1I and J*). Furthermore, the levels of LAST and c-Myc appeared to be notably synchronous during cell cycle progression (*Figure 1—figure supplement 1C*). In particular, the c-Myc and *LAST* levels were decreased in G2/M (lane 2), followed by a rapid increase in the G1 phase (lane 3). These data suggest that *LAST* expression is positively regulated by c-Myc.

We next explored whether c-Myc regulates *LAST* expression at the transcriptional level. We first inspected the genomic sequence around the *LAST* gene using the JASPAR database (*Mathelier et al., 2016*). Six putative c-Myc binding sites (BS1, BS2, BS3, BS4, BS5 and BS6) were identified (*Figure 1K*, upper panel). Furthermore, we analyzed the genomic sequence around the *LAST* gene using the ENCODE database. Three fragments (F1, F2 and F3) were predicted to be

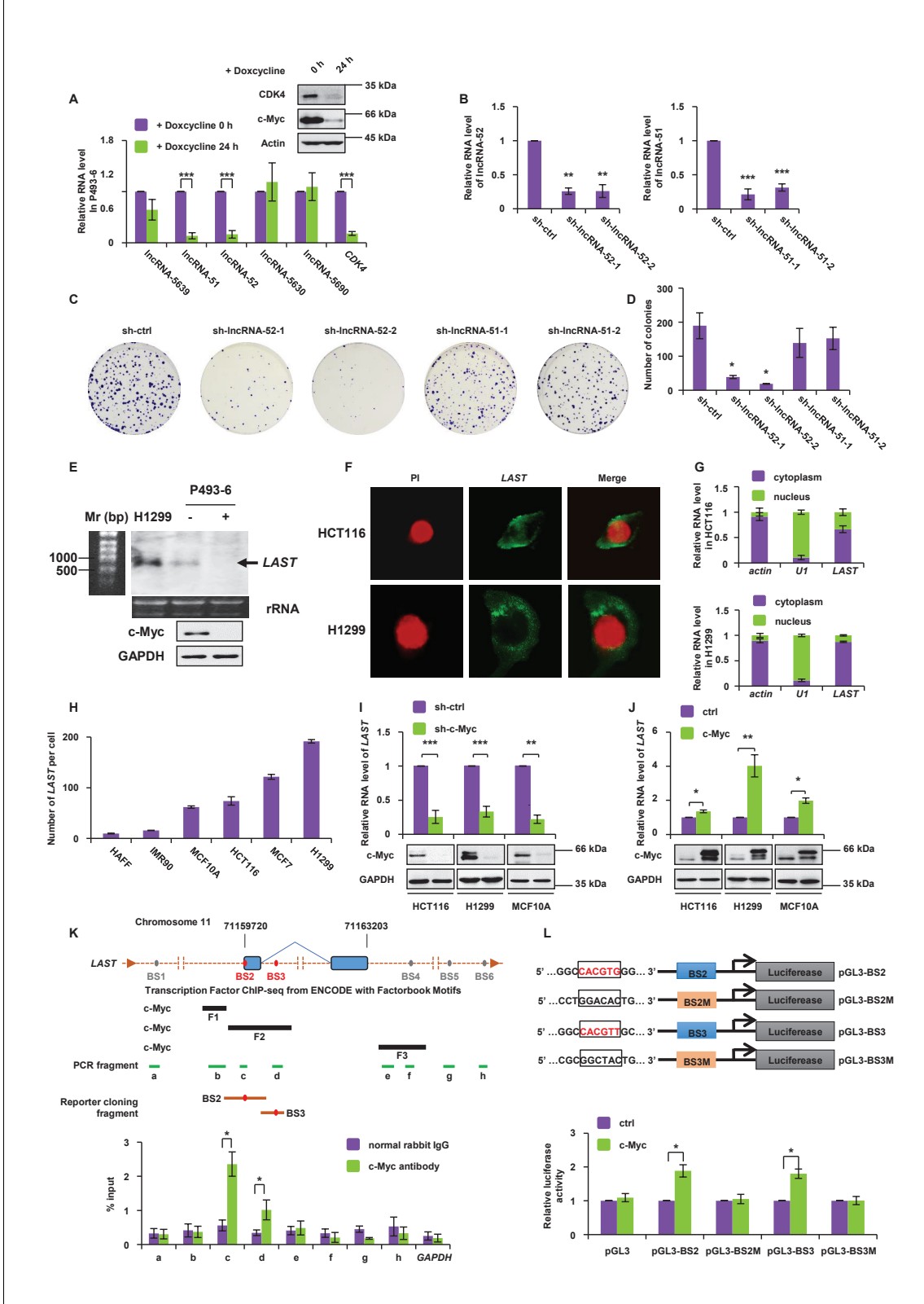

**Figure 1.** *LAST* is positively regulated by c-Myc. (**A**) P493-6 cells carrying a c-Myc tet-off system were treated with doxycycline (1 μg/mL) for 24 hr. The levels of five lncRNAs (lncRNA-5639,–51, −52,–5630 and −5690) and the positive control *CDK4* were assessed by real-time RT–PCR analysis. Data shown are the mean ± SD (n = 3; ***p<0.001, two-tailed t-test). Cell lysates were also analyzed by western blotting with the indicated antibodies to ensure that gene expression was controlled by c-Myc. (**B**) HCT116 cells were infected with lentiviruses expressing control shRNA (sh-ctrl), shRNA-1,–2 against

*Figure 1 continued on next page*

*Figure 1 continued*

lncRNA-52 or shRNA-1,–2 against lncRNA-51, as indicated. The lentivirus-mediated gene knockdown efficiencies for both lncRNA-52 and lncRNA-51 were analyzed by real-time RT–PCR. Data shown are the mean ± SD (n = 3; **p<0.01, ***p<0.001, two-tailed t-test). (C) Colonies of the above cells were stained with crystal violet and photographed after 14 days of incubation. (D) The number of colonies was counted and plotted in columns. (E) Total RNA from the indicated cell lines was subjected to northern blot analysis to determine the molecular size of *LAST*. (F) Single molecule RNA FISH detecting endogenous *LAST* molecules (green) in HCT116 and H1299. Chromosomal DNA (red) was stained with PI. (G) HCT116 and H1299 cells were fractionated into cytoplasmic and nuclear extracts. Total RNA extracted from each fraction was analyzed by real-time RT–PCR. Data shown are the mean ± SD (n = 3). Actin and U1 were used as markers for the cytoplasmic and nuclear fractions, respectively. (H) The *LAST* transcript copy numbers per cell in HAFF, IMR90, MCF10A, HCT116, MCF7 and H1299 cells were determined by absolute quantitative PCR (qPCR). Data shown are the mean ± SD (n = 3). (I) HCT116, H1299 and MCF10A cells were infected with lentiviruses expressing control shRNA or c-Myc shRNA. Ninety-six hours after infection, total RNA and cell lysates were analyzed by real-time RT-PCR and western blotting, respectively. Data shown are the mean ± SD (n = 3; **p<0.01, ***p<0.001, two-tailed t-test). (J) HCT116, H1299 and MCF10A cells were transfected with empty vector or FLAG-c-Myc. Twenty-four hours after transfection, total RNA was extracted from these cells and subjected to real-time RT-PCR analysis. Data shown are the mean ± SD (n = 3; *p<0.05, **p<0.01, two-tailed t-test). Cell lysates were also analyzed by western blotting using the indicated antibodies. (K) Schematic representation of putative c-Myc binding sites around the *LAST* gene, predicted c-Myc binding fragments, qPCR-amplified fragments from the ChIP assay and fragments used in the luciferase reporter assay (upper panel). Lysates from HCT116 cells were subjected to the ChIP assay with a normal rabbit IgG or c-Myc antibody. ChIP products were amplified by qPCR with the indicated pairs of primers (*Table 1*). Data shown are the mean ± SD (n = 3; *p<0.05, two-tailed t-test) (lower panel). (L) Schematic diagram of the luciferase reporter systems constructed to assess *LAST* promoter activity. The indicated pGL3-based luciferase reporter constructs were generated to examine the transcriptional activities of two putative c-Myc binding sites, BS2 and BS3, in response to c-Myc induction. BS2M and BS3M indicate their corresponding mutant binding sites, which are written in black in the open boxes (upper panel). HCT116 cells were co-transfected with either FLAG-c-Myc or the control vector plus the indicated reporter constructs and Renilla luciferase plasmid. Twenty-four hours after transfection, reporter activity was measured and plotted after normalizing with respect to Renilla luciferase activity. Data shown are the mean ± SD (n = 3; *p<0.05, two-tailed t-test) (lower panel).

DOI: https://doi.org/10.7554/eLife.30433.003

The following source data and figure supplements are available for figure 1:

**Source data 1.** Source data for *Figure 1A, B, D, G, H, I, J, K and L*.
DOI: https://doi.org/10.7554/eLife.30433.006
**Figure supplement 1.** LAST is positively regulated by c-Myc.
DOI: https://doi.org/10.7554/eLife.30433.004
**Figure supplement 2.** Uncropped images of blots.
DOI: https://doi.org/10.7554/eLife.30433.005

recognized by c-Myc (*Figure 1K*, upper panel). The chromatin immunoprecipitation (ChIP) assay determined the association of c-Myc with chromatin fragments corresponding to the BS2 and BS3 sites (within F2 fragment) among all examined fragments (*Figure 1K*, lower panel). We further evaluated whether BS2 and BS3 conferred c-Myc-dependent transcriptional activity. DNA fragments containing wild-type BS2 and BS3 or their corresponding mutant binding sites were inserted into the promoter region of a firefly luciferase reporter plasmid (*Figure 1L*, upper panel). Luciferase expression from the reporter containing an individual BS2 or BS3 site was indeed induced by ectopic expression of c-Myc (*Figure 1L*, lower panel). By contrast, mutant BS2M and BS3M sites showed no response to c-Myc induction (*Figure 1L*, lower panel). These data demonstrate that c-Myc transactivates *LAST*.

### *LAST* promotes G1/S transition and upregulates *cyclin D1/CCND1*

Knockdown of *LAST* results in reduced colony formation (*Figure 1C*), indicating that *LAST* normally promotes cell proliferation. To examine how *LAST* affects cell growth, the *cell cycle phase* distribution was analyzed by *flow* cytometric *analysis*. Knockdown of *LAST* caused a decrease in the percentage of cells in the S and G2/M phases and an increase in the percentage of cells in the G1 phase (*Figure 2—figure supplement 1A and B*), indicating that *LAST* knockdown prevents cell passage from the G1 phase into S phase. As a result, *LAST* was shown to promote G1/S phase transition.

Cell cycle regulation is controlled by many factors. To define which factor(s) were involved in *LAST*-mediated regulation, the mRNA levels of G1-related cyclins and CDKs genes were selected for comparison in HCT116 cells before and after *LAST* gene knockdown. Among all of the mRNAs examined, only the *CCND1* mRNA level was significantly decreased (*Figure 2A*). Among all of the cyclins or CDKs examined, only cyclin D1 was shown to be downregulated when *LAST* was depleted (*Figure 2B*). The lncRNA *PVT1* is known to be a c-Myc regulated lncRNA that is involved in c-Myc

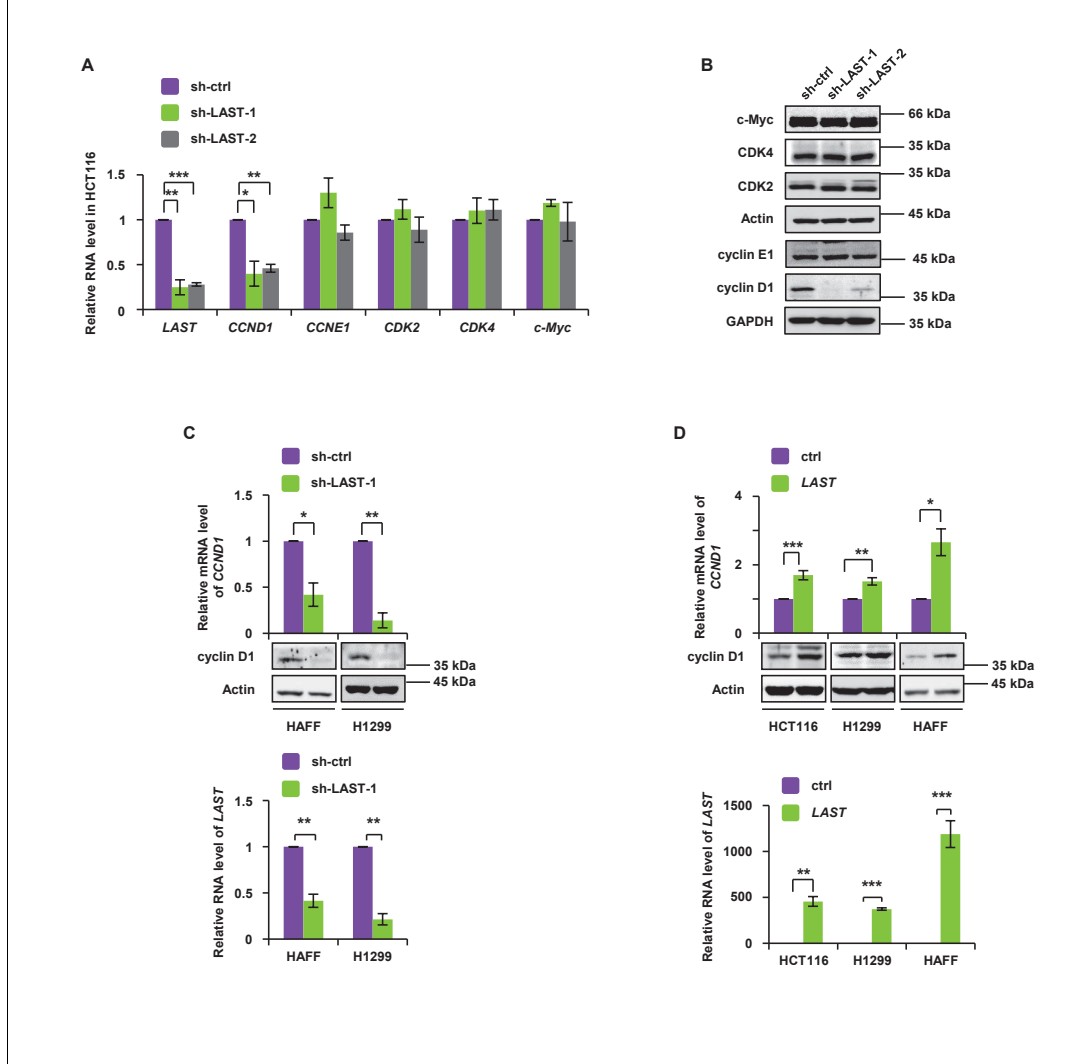

**Figure 2.** *LAST* accelerates G1/S transition and upregulates *cyclin D1/CCND1*. (**A**) HCT116 cells were infected with lentiviruses expressing control shRNA, *LAST* shRNA-1 or −2. Ninety-six hours after infection, total RNA was extracted and the transcript levels for *LAST*, *CCND1*, *CCNE1*, *CDK2*, *CDK4* and *c-Myc* were analyzed by real-time RT-PCR. Data shown are the mean ± SD (n = 3; *p<0.05, **p<0.01, ***p<0.001, two-tailed t-test). (**B**) Cell lysates prepared as described above (*Figure 2A*) were analyzed by western blotting to examine GAPDH, cyclin D1, cyclin E1, Actin, CDK2, CDK4 and c-Myc expression. (**C**) HAFF and H1299 cells were infected with lentiviruses expressing control shRNA or *LAST* shRNA. Ninety-six hours after infection, total RNA was analyzed by real-time RT-PCR to detect the level of *LAST* to determine its knockdown efficiency (lower panel). Total RNA was also analyzed by real-time RT-PCR to detect the level of *CCND1* mRNA and by western blotting to examine the cyclin D1 protein level (upper panel). Data shown are the mean ± SD (n = 3; *p<0.05, **p<0.01). (**D**) HCT116, H1299 and HAFF cells were infected with lentiviruses expressing control RNA or *LAST*. Ninety-six hours after infection, total RNA was analyzed by real-time RT-PCR to detect successful expression of *LAST* (lower panel). Total RNA was also analyzed by real-time RT-PCR to detect the level of *CCND1* mRNA and by western blotting to examine the cyclin D1 protein level (upper panel). Data shown are the mean ± SD (n = 3; *p<0.05, **p<0.01, ***p<0.001, two-tailed t-test).

DOI: https://doi.org/10.7554/eLife.30433.007

The following source data and figure supplements are available for figure 2:

**Source data 1.** Source data for *Figure 2A, C and D*.
DOI: https://doi.org/10.7554/eLife.30433.010
**Figure supplement 1.** LAST knockdown prevents cell passage from the G1 phase into S phase.
DOI: https://doi.org/10.7554/eLife.30433.008
**Figure supplement 2.** Uncropped images of blots.
DOI: https://doi.org/10.7554/eLife.30433.009

stability and activity (*Colombo et al., 2015*). However, unlike *PVT1*, we found that *LAST,* which is also regulated by c-Myc (*Figure 1A*), does not affect c-Myc stability since knockdown of *LAST* did not change c-Myc expression at either the mRNA or protein levels (*Figure 2A and B*). The effect of *LAST* on cyclin D1/*CCND1* was further verified in normal HAFF cells and tumor H1299 and HCT116 cells. Depletion of *LAST* decreased whereas over-expression of *LAST* increased cyclin D1/*CCND1* expression at both the mRNA and protein levels (*Figure 2C and D*).

To test if the function of LAST is mediated through an effect on the adjacent genes, we checked the shRNA-mediated *LAST* knockdown effect on the two adjacent genes *DHCR7* and *NADSYN1* (*Figure 1—figure supplement 1A*) and found that *LAST* showed no effect on either the mRNA or protein levels of those two genes (*Figure 2—figure supplement 1C*). Furthermore, we introduced shRNA-resistant *LAST* into *LAST* depleted cells, and as shown in *Figure 2—figure supplement 1D and E*, both the *CCND1* mRNA and protein levels were rescued. This result excludes off-target effects of *LAST* shRNA knockdown.

## *LAST* cooperates with CNBP to regulate *CCND1* mRNA stability

To investigate how *LAST* affects the *CCND1* mRNA level, we first examined whether *LAST* regulates the *CCND1* mRNA transcription process. The levels of both *CCND1* pre-mRNA and mature mRNA were examined by primers against *CCND1* mRNA intron- or exon-specific regions in HCT116 cells treated with and without *LAST* knockdown. The levels of *CCND1* pre-mRNA containing four intronic regions were found to remain unaltered between control and *LAST* knockdown cells, whereas the levels of mature spliced *CCND1* mRNA containing 5'UTR, CDS (coding sequences) and 3'UTR regions were greatly reduced upon *LAST* depletion (*Figure 3—figure supplement 1A and B*). These results suggest that *LAST* may post-transcriptionally regulate *CCND1* mRNA. To evaluate the effect of *LAST* on the stability of *CCND1* mRNA, HCT116 cells were treated with actinomycin D, which measures the decay of pre-existing mRNA. Knockdown of *LAST* resulted in a decrease of the half-life of *CCND1* mRNA from 5 hr to 3 hr (*Figure 3A*), whereas over-expression of *LAST* increased its half-life from 5 hr to 9 hr (*Figure 3B*), indicating that *LAST* stabilizes *CCND1* mRNA. To determine whether *LAST* interacts with *CCND1* mRNA, we performed a biotinylated oligonucleotide pull-down assay, and as shown in *Figure 3C*, endogenous *CCND1* mRNA but not *CCNB1* mRNA co-precipitated with *LAST*, indicating an association between *LAST* and *CCND1* mRNA. However, by careful inspection, we found there was no complementary base pairing between *LAST* and *CCND1* mRNA. We therefore hypothesized that some protein(s) may mediate this binding. Proteins pulled down by *LAST* were separated by SDS PAGE, and a unique band with a molecular weight of approximately 20 kDa was revealed and identified as CNBP by mass spectrometry (*Figure 3—figure supplement 1C*, left panel). CNBP has a preference for binding single-stranded DNA and RNA (*Flink and Morkin, 1995*) and has been reported to function in the translation of ornithine decarboxylase mRNA (*Sammons et al., 2010*). To validate the MS Spectro result, we performed a *LAST* pull-down assay. A biotin-labeled antisense DNA probe against *LAST* pulled down CNBP, but not cyclin D1 (*Figure 3—figure supplement 1C*, right panel). To further demonstrate that CNBP can bridge *CCND1* mRNA and *LAST*, we first pulled down *CCND1* mRNA using a biotinylated antisense DNA probe as bait; both CNBP and *LAST* were coprecipitated (*Figure 3D*). The RIP assay further concluded that CNBP interacts with both *CCND1* mRNA and *LAST* (*Figure 3E*). These data demonstrate that CNBP acts as a mediator for *LAST* and *CCND1* mRNA binding. As shown in *Figure 3F*, CNBP knockdown in HCT116 led to a decrease in both the mRNA and protein levels of *cyclin D1/CCND1*. The effect of CNBP on the stability of *CCND1* mRNA was evaluated in HCT116 cells treated with actinomycin D. The half-life of *CCND1* mRNA was reduced from 5 hr to 3 hr as CNBP was depleted (*Figure 3G*), further demonstrating that CNBP prolongs the *CCND1* mRNA half-life. Because CNBP predominantly resides in the cytosol, we investigated whether the association of *CCND1* mRNA with *LAST* via CNBP only occurs in the cytosol. Using a *LAST* and *CCND1* mRNA pull-down assay, we found that CNBP was co-precipitated by either *LAST* or *CCND1* mRNA in the cytoplasm, but not the nucleus (*Figure 3—figure supplement 1D*). HNRNPK was used as a nuclear marker. Moreover, knockdown of CNBP was shown to result in decreased levels of *CCND1* mRNA (*Figure 3F*). Further investigation revealed that knockdown of CNBP affected the level of mature *CCND1* mRNA, but not unspliced *CCND1* pre-mRNA, indicating that the protective role of CNBP in mature *CCND1* mRNA stability occurred in the cytosol since nuclear unspliced *CCND1* pre-mRNA was not affected when CNBP was silenced (*Figure 3—figure supplement 1E*). To further confirm that *LAST* affects *CCND1* mRNA

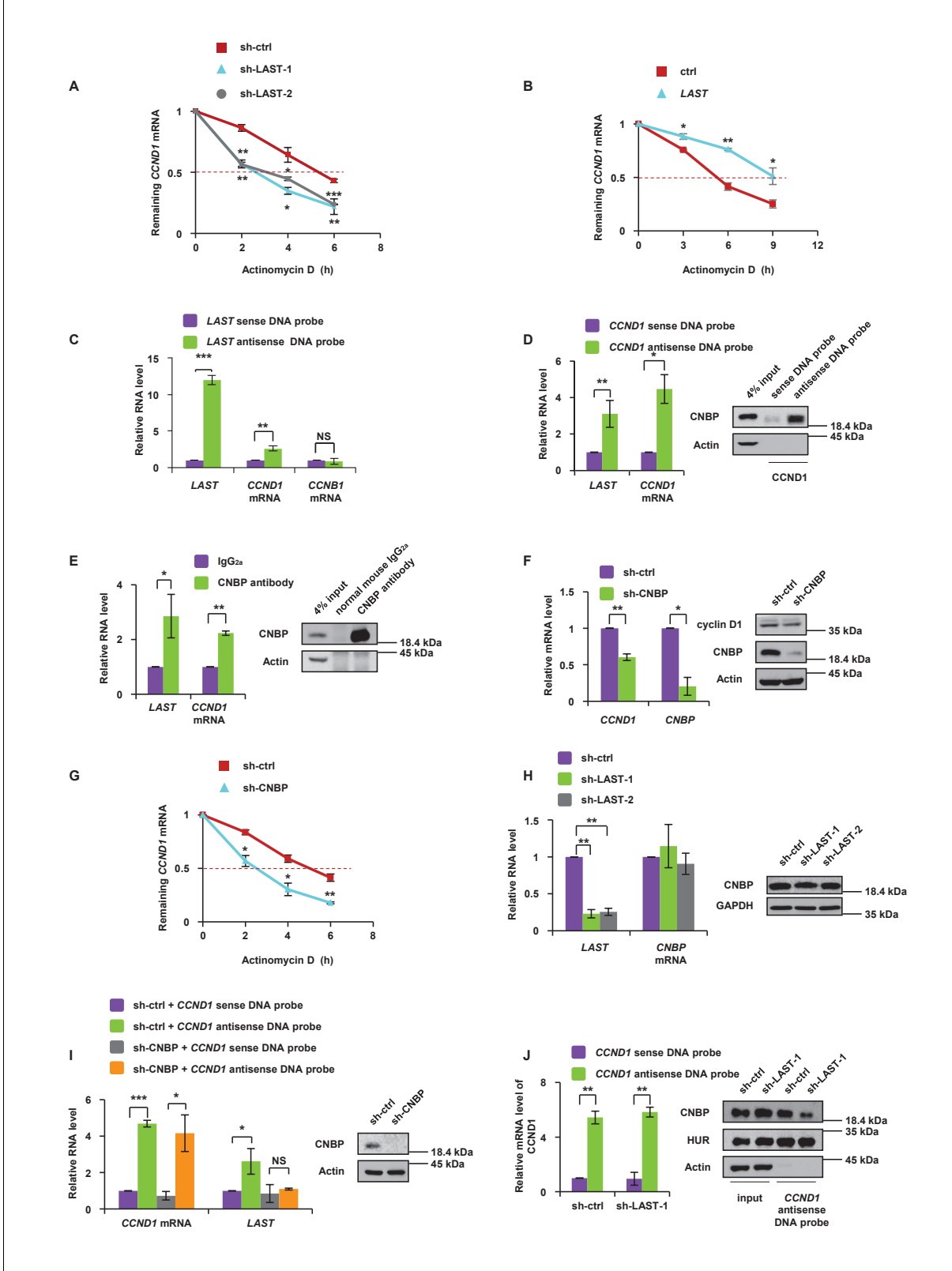

**Figure 3.** *LAST* stabilizes *CCND1* mRNA via CNBP. (**A**) HCT116 cells expressing control shRNA, *LAST* shRNA-1 or −2 were treated with actinomycin D (1 µg/mL) for the indicated periods of time. Total RNA was purified and then analyzed by real-time RT-PCR to examine the mRNA half-life of *CCND1*. Data shown are the mean ± SD (n = 3; *p<0.05, **p<0.01, ***p<0.001, two-tailed t-test). (**B**) HCT116 cells expressing control RNA or *LAST* were treated with actinomycin D (1 µg/mL) for the indicated periods of time. Total RNA was extracted and then analyzed by real-time RT-PCR to examine the mRNA
*Figure 3 continued on next page*

*Figure 3 continued*

half-life of *CCND1*. Data shown are the mean ± SD (n = 3; *p<0.05, **p<0.01, two-tailed t-test). (C) HCT116 cell lysates were incubated with in vitro synthesized biotin-labeled sense or antisense DNA probes against *LAST* for the biotinylated oligonucleotide pull-down assay. The precipitates from the pull-down were analyzed by real-time RT-PCR to detect the interacting mRNAs. Data shown are the mean ± SD (n = 3; **p<0.01, ***p<0.001, two-tailed t-test). (D) HCT116 cell lysates were incubated with in vitro synthesized biotin-labeled sense or antisense DNA probes against *CCND1* mRNA for the biotin pull-down assay. The precipitates from the pull-down underwent real-time RT-PCR and western blot analyses to examine the levels of indicated RNAs and protein CNBP, respectively. Data shown are the mean ± SD (n = 3; *p<0.05, **p<0.01, two-tailed t-test). (E) HCT116 cell lysates were immunoprecipitated with an antibody against CNBP or normal mouse IgG$_{2a}$. Precipitated samples were analyzed by western blotting to ensure successful precipitation of CNBP and by real-time RT-PCR to detect the indicated coprecipitated RNAs. Data shown are the mean ± SD (n = 3; *p<0.05, **p<0.01, two-tailed t-test). (F) HCT116 cells were infected with lentiviruses expressing control shRNA or *CNBP* shRNA. Ninety-six hours after infection, total RNA was subjected to real-time RT-PCR to compare the levels of *CCND1* mRNA. Data shown are the mean ± SD (n = 3; *p<0.05, **p<0.01, two-tailed t-test). Cell lysates were also analyzed by western blotting with the indicated antibodies. (G) HCT116 cells expressing control shRNA or *CNBP* shRNA were treated with actinomycin D (1 µg/mL) for the indicated periods of time. Total RNA was then analyzed by real-time RT-PCR to examine the mRNA half-life of *CCND1*. Data shown are the mean ± SD (n = 3; *p<0.05, **p<0.01, two-tailed t-test). (H) HCT116 cells were infected with lentiviruses expressing control shRNA, *LAST* shRNA-1 or −2. Ninety-six hours after infection, total RNA was subjected to real-time RT-PCR to compare the levels of *CNBP*. Data shown are the mean ± SD (n = 3; **p<0.01, two-tailed t-test). Cell lysates were also analyzed by western blotting to examine actin and CNBP expression. (I) Cell lysates of HCT116 cells expressing control shRNA or *CNBP* shRNA were incubated separately with in vitro synthesized biotin-labeled sense or antisense DNA probes against *CCND1* mRNA for the biotinylated oligonucleotide pull-down assay. The pull-down products were subjected to real-time RT-PCR analysis to examine the indicated RNA levels. Cell lysates from HCT116 treated with or without *CNBP* shRNA knockdown were subjected to western blotting to ensure knockdown of CNBP. Data shown are the mean ± SD (n = 3; *p<0.05, ***p<0.001, two-tailed t-test). (J) Cell lysates of HCT116 cells expressing control shRNA or *LAST* shRNA-1 were incubated with in vitro synthesized biotin-labeled sense or antisense DNA probes against *CCND1* mRNA for the biotin pull-down assay, followed by real-time RT-PCR analysis to examine the indicated RNA levels. Pull-down products were also subjected to western blotting with the indicated antibodies and real-time RT-PCR. Data shown are the mean ± SD (n = 3; **p<0.01, two-tailed t-test).

DOI: https://doi.org/10.7554/eLife.30433.011

The following source data and figure supplements are available for figure 3:

**Source data 1.** Source data for *Figure 3A, B, C, D, E, F, G, H, I and J*.
DOI: https://doi.org/10.7554/eLife.30433.014
**Figure supplement 1.** LAST stabilizes CCND1 mRNA via CNBP.
DOI: https://doi.org/10.7554/eLife.30433.012
**Figure supplement 2.** Uncropped images of blots.
DOI: https://doi.org/10.7554/eLife.30433.013

stability through CNBP, we knocked down CNBP in HCT116 cells. As shown in *Figure 3—figure supplement 1F*, the increased expression of *CCND1* mRNA caused by over-expression of *LAST* was diminished when CNBP was depleted (lanes 2 vs. 4). Thus, our hypothesis is that *LAST* affects *CCND1* mRNA via CNBP. Knockdown of *LAST* resulted in no change in CNBP at either the RNA or protein levels (*Figure 3H*), which suggests that *LAST* affects *CCND1* mRNA not according to the quantity of CNBP, but rather by the association of CNBP and *CCND1* mRNA. It was therefore expected that knockdown of CNBP would reduce the association between *LAST* and *CCND1* mRNA. This was indeed the case. An RNA pull-down experiment was performed starting with the same amount of *CCND1* mRNA, and less *LAST* was co-precipitated as CNBP was depleted (*Figure 3I*). Similarly, when we pulled down the same amount of *CCND1* mRNA, less co-precipitated CNBP was detected as *LAST* was silenced. As a negative control, the RNA-binding protein HuR remained unchanged after *LAST* was knocked down (*Figure 3J*). These data suggest that *LAST* cooperates with CNBP to regulate *CCND1* mRNA stability.

## Both *LAST* and *CCND1* mRNA bind to CNBP through their G-rich motifs

To describe a detailed CNBP, *LAST* and *CCND1* mRNA binding mechanism, we mapped the *LAST* and CNBP binding sites on *CCND1* mRNA by RNA pull-down using different in vitro biotin-labeled fragments (*Figure 4A*, upper panel). We found that the 5'UTR but not 3'UTR-1,–2 and −3 of *CCND1* mRNA was able to bind *LAST* and CNBP (*Figure 4A and B*), indicating that *LAST* and CNBP bind to the 5' region of *CCND1* mRNA. To further determine whether *LAST* and CNBP bind to the *CCND1* mRNA 5'UTR to enhance its stability, two *CCND1* expression constructs were generated, as shown in *Figure 4—figure supplement 1A and B*. One construct contained the *CCND1* coding region (CD) plus the 5'UTR and the other contained the CD alone. The expression plasmid plus *LAST* and

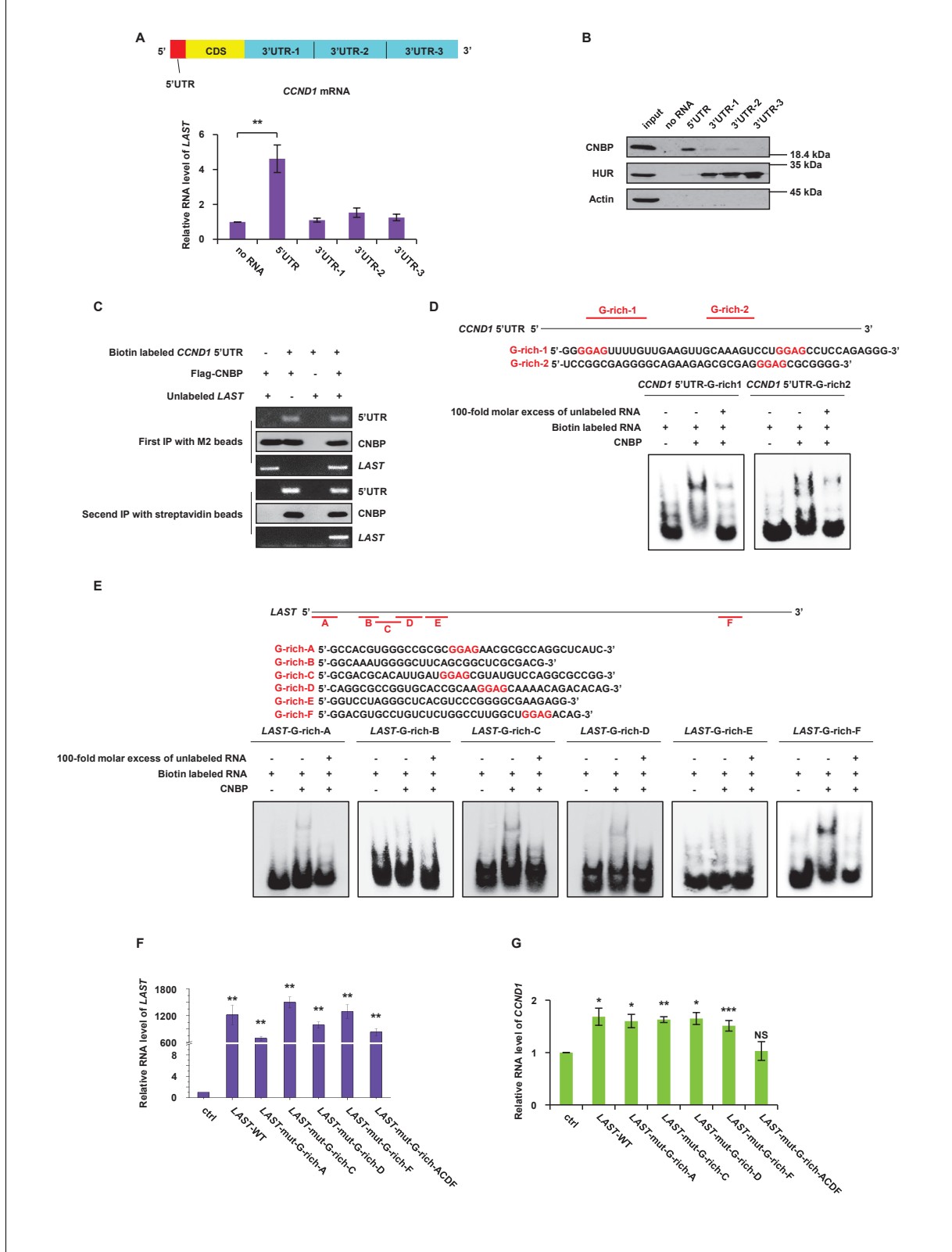

**Figure 4.** CNBP binds to *LAST* and *CCND1* mRNA via their G-rich motifs. (**A**) Schematic illustration showing different parts, including the 5'UTR, CDS, 3'UTR-1, 3'UTR-2 and 3'UTR-3, in *CCND1* mRNA (upper panel). HCT116 cell lysates were incubated with in vitro synthesized biotin-labeled *CCND1* 5'UTR as well as 3'UTR-1,−2, and −3 (upper panel), followed by RNA pull-down. Cell lysates incubated with no RNA were used as negative controls. Pull-down products were subjected to real-time RT-PCR. Data shown are the mean ± SD (n = 3; **p<0.01, two-tailed t-test) (lower panel). (**B**) The pull-

*Figure 4 continued on next page*

*Figure 4 continued*

down products from above were analyzed by western blotting with the indicated antibodies. (C) In-vitro synthetic biotin-labeled *CCND1* 5'UTR and unlabeled *LAST* plus Flag-CNBP were incubated for 3 hr at 4°C. The mixtures were first immunoprecipitated with M2 beads, followed by elution with 3 × FLAG peptides. Ten percent of the eluent was analyzed by western blotting or RT-PCR. The rest of the eluent was further immunoprecipitated with streptavidin beads. The immunoprecipitates were then washed. After elution, 10% of the eluent was analyzed by western blotting. Ninety percent of the eluent was used for real-time RT-PCR analysis. (D) A schematic illustration of two G-rich regions in the *CCND1* 5'UTR. Electrophoretic mobility shift assay (EMSA) was performed to detect the CNBP binding activity to its targeted G-rich region 1 and 2. (E) A schematic illustration of six G-rich regions in *LAST*. An electrophoretic mobility shift assay (EMSA) was performed to detect the CNBP binding activity to its targeted G-rich motif A, B, C, D, E and F. (F) HCT116 cells were infected with lentiviruses expressing either control RNA; wild-type *LAST*; *LAST* individually mutated at G-rich-A, G-rich-C, G-rich-D and G-rich-F sites; or LAST mutated at the four G-rich-A, C, D and F sites combined. Ninety-six hours after infection, total RNA was analyzed by real-time RT-PCR to detect the successful expression levels of *LAST* or mutant *LAST*. Data shown are the mean ± SD (n = 3; **p<0.01, two-tailed t-test). (G) Total RNA of HCT116 cells separately expressing exogenous control RNA; wild-type *LAST*; *LAST* with a single mutation at G-rich-A, G-rich-C, G-rich-D or G-rich-D; and *LAST* with four mutations combined were analyzed by real-time RT-PCR to detect the mRNA level of *CCND1*. Data shown are the mean ± SD (n = 3; *p<0.05, **p<0.01, ***p<0.001, two-tailed t-test).

DOI: https://doi.org/10.7554/eLife.30433.015

The following source data and figure supplements are available for figure 4:

**Source data 1.** Source data for *Figure 4A, F and G*.
DOI: https://doi.org/10.7554/eLife.30433.019
**Figure supplement 1.** CNBP binds to CCND1 mRNA 5'UTR.
DOI: https://doi.org/10.7554/eLife.30433.016
**Figure supplement 2.** Both LAST and CCND1 mRNA bind to CNBP.
DOI: https://doi.org/10.7554/eLife.30433.017
**Figure supplement 3.** Uncropped images of blots.
DOI: https://doi.org/10.7554/eLife.30433.018

CNBP or plasmid alone was individually transfected into 293T cells. Actinomycin D was added to measure the mRNA decay rate. The half-life of ectopically expressed *CCND1* mRNA lacking the 5'UTR was not altered by the presence or absence of *LAST* and CNBP (*Figure 4—figure supplement 1A*). By contrast, the half-life of *CCND1* mRNA bearing 5'-UTRs was extended from 4 hr in the absence of *LAST* and CNBP to 9 hr in the presence of *LAST* and CNBP, indicating that *LAST* and CNBP enhanced *CCND1* mRNA stability via its 5'UTR (*Figure 4—figure supplement 1B*). In addition, we performed CNBP RIP sequencing, and an enrichment peak at the *CCND1* 5'UTR was observed (*Figure 4—figure supplement 1C*). This was consistent with the previous conclusion from *Figure 4B*. Thus, our hypothesis is that CNBP binds both *CCND1* mRNA and *LAST*. We again examined whether *LAST*, the 5'-UTR of *CCND1* mRNA and CNBP form a ternary complex by using a sequential immuno-precipitation assay (*Figure 4—figure supplement 2A*). By using an anti-FLAG antibody against FLAG–CNBP, both the biotin-labeled 5'-UTR of *CCND1* mRNA and *LAST* were pulled down in an initial immunoprecipitation assay (*Figure 4C*, panel 1 and 3). The immunocomplexes were eluted and were subsequently precipitated by streptavidin sepharose beads against the biotin-labeled 5'-UTR of *CCND1* mRNA. *LAST* and CNBP were present in the streptavidin-biotin precipitates (*Figure 4C*, panel 5 and 6), indicating that these three components indeed form a ternary complex.

CNBP prefers to bind G-rich motifs, especially the GGAG core (*Armas et al., 2008*; *Benhalevy et al., 2017*). We checked the proportion of G-rich motifs in all of the peak sequences from the CNBP RIP samples. Nearly sixty percent of the CNBP enriched sequences contained the GGAG motif, and more than ninety percent of the peak sequences contained a GGR motif (*Figure 4—figure supplement 2B*). To assess the possible CNBP binding sites on *LAST* and *CCND1* mRNA, the bioinformatics software tool QGRS Mapper was utilized (*Kikin et al., 2006*). Two G-rich sequences containing a GGAG core in the 5'UTR of *CCND1* mRNA were identified (*Figure 4D*, upper part). An electrophoretic mobility shift assay (EMSA) was performed, and the results showed that G-rich-1 and G-rich-2 in the *CCND1* 5'UTR were responsible for the binding of CNBP (*Figure 4D*, lower part). Among the six predicted G-rich sequences (G-rich-A to F) found in *LAST* (*Figure 4E*, upper part), four G-rich sequences (G-rich-A, C, D and F) were found to contain a GGAG core. G-rich-A, C, D and F from *LAST* were able to bind CNBP, whereas neither G-rich-B nor G-rich-E was able to bind CNBP (*Figure 4E*, lower part). Thus, CNBP only interacted with G-rich sequences that contained the GGAG core, but not those lacking the GGAG core. These data

suggest that both *CCND1* and *LAST* interact with CNBP via their G-rich motifs containing the GGAG core. Four G-rich regions (A, C, D and F) were mutated from GGAG to UUUU with either a single mutation or four combined mutations in *LAST*. We found that over-expression of *LAST* containing only one site mutation led to an increase in the *CCND1* mRNA level, whereas over-expression of *LAST* containing four G-rich site mutations nullified its effect on the *CCND1* mRNA level (*Figure 4F and G*). This result indicates that the effect of *LAST* on *CCND1* stability requires at least one of the four functional G-rich motifs. To define which domain of CNBP is responsible for binding *LAST* and the *CCND1* 5'UTR, a biotin-labeled RNA pull-down assay and deletion mapping were performed. According to the web site InterPro (*Hunter et al., 2009*), CNBP can be divided into four structural domains based on its zinc-finger arrangement (*Figure 4—figure supplement 2C*). As shown in *Figure 4—figure supplement 2C*, we concluded that *LAST* binds to the CNBP fragment corresponding to amino acids 92–134 (domain 3), whereas the *CCND1* 5'UTR binds to the CNBP fragment corresponding to amino acids 29–134 (domain 2 + domain 3) (*Figure 4—figure supplement 2C*). Determination of the exact mechanism of these associations requires further investigation.

## In addition to *CCND1*, *LAST* regulates the stability of other mRNAs

To globally identify transcripts that simultaneously meet the following requirements: (i) transcripts are downregulated by *LAST* knockdown and (ii) transcripts are able to bind to CNBP, we assembled two unbiased transcriptome profiles using *LAST* knockdown mRNA-seq and CNBP RIP-seq in HCT116 cells. The intersection of these two arrays is shown in *Figure 5A*, and 225 overlapping genes were found (*Supplementary file 2*). We further narrowed this list down to 75 genes (*Supplementary file 2*, bold part) based on the criteria that CNBP-enriched genes must be 4-fold above the input control level. Three mRNAs, namely, *SOX9*, *NFE2L1* and *PDF,* were also likely to be regulated by *LAST*, as knockdown of *LAST* led to a decrease in their levels (*Figure 5B*). Experimental verification showed that knockdown of *LAST* decreased (*Figure 5C*) whereas over-expression of *LAST* increased their half-lives (*Figure 5D*). In addition, CNBP deletion led to a decrease in the mRNA levels of *SOX9*, *NFE2L1* and *PDF* (*Figure 5E and F*). These data suggest that *LAST*, together with CNBP, can regulate the stabilization of additional mRNAs, such as *SOX9*, *NFE2L1* and *PDF*.

## *LAST* promotes tumorigenesis

To further determine whether *LAST* regulates tumorigenesis, we used a xenograft mouse model. HCT116 cells stably expressing exogenous *LAST* or *LAST* shRNA-1 were injected subcutaneously into the dorsal flanks of nude mice (left (control) and right (treated), n = 7 for each group). According to animal care and enforcement, mice were sacrificed when the largest subcutaneous tumor mass on one flank was close to one cubic centimeter. Tumors expressing control shRNA or *LAST* shRNA-1 were excised after 6 weeks, and tumors expressing control RNA or *LAST* were excised after 3 weeks. Mice were sacrificed and tumors were excised. Knockdown of *LAST* decreased the tumorigenicity of HCT116 cells (*Figure 6A and B*). By contrast, induction of *LAST* promoted HCT116 cell tumorigenicity (*Figure 6C and D*). Furthermore, based on the TCGA dataset (*Weinstein et al., 2013*), we found that the *LAST* expression levels were higher in tumor tissues than normal tissues, including the human bladder, breast, colorectal, esophagus, head and neck, kidney, liver, lung, prostate and stomach. In addition, the *CCND1* expression levels were higher in tumor tissues than in normal tissues, including the human bladder, breast, cervix, bile duct, colorectal, esophagus, head and neck, kidney, pancreas, stomach and uterus. In conclusion, both the *LAST* and *CCND1* expression levels were higher in most tumor tissues than in their normal counterparts (*Figure 6E and F*, *Figure 6—figure supplements 1* and *2*). The above results suggest that *LAST* promotes tumorigenesis.

To assess the impact of *LAST* deficiency on gene expression in HTC116, we performed unbiased transcriptome profiling using RNA-seq in HCT116 cells. The absence of *LAST* downregulated expression of 667 genes (log2 (fold change) below - 0.58) (*Supplementary file 3*). We then performed pathway analysis in those genes and found the top 10 significant pathways that were significantly associated with 667 differentially expressed genes. Among these 10 pathways, the majority were associated with tumorigenesis (*Figure 6G*).

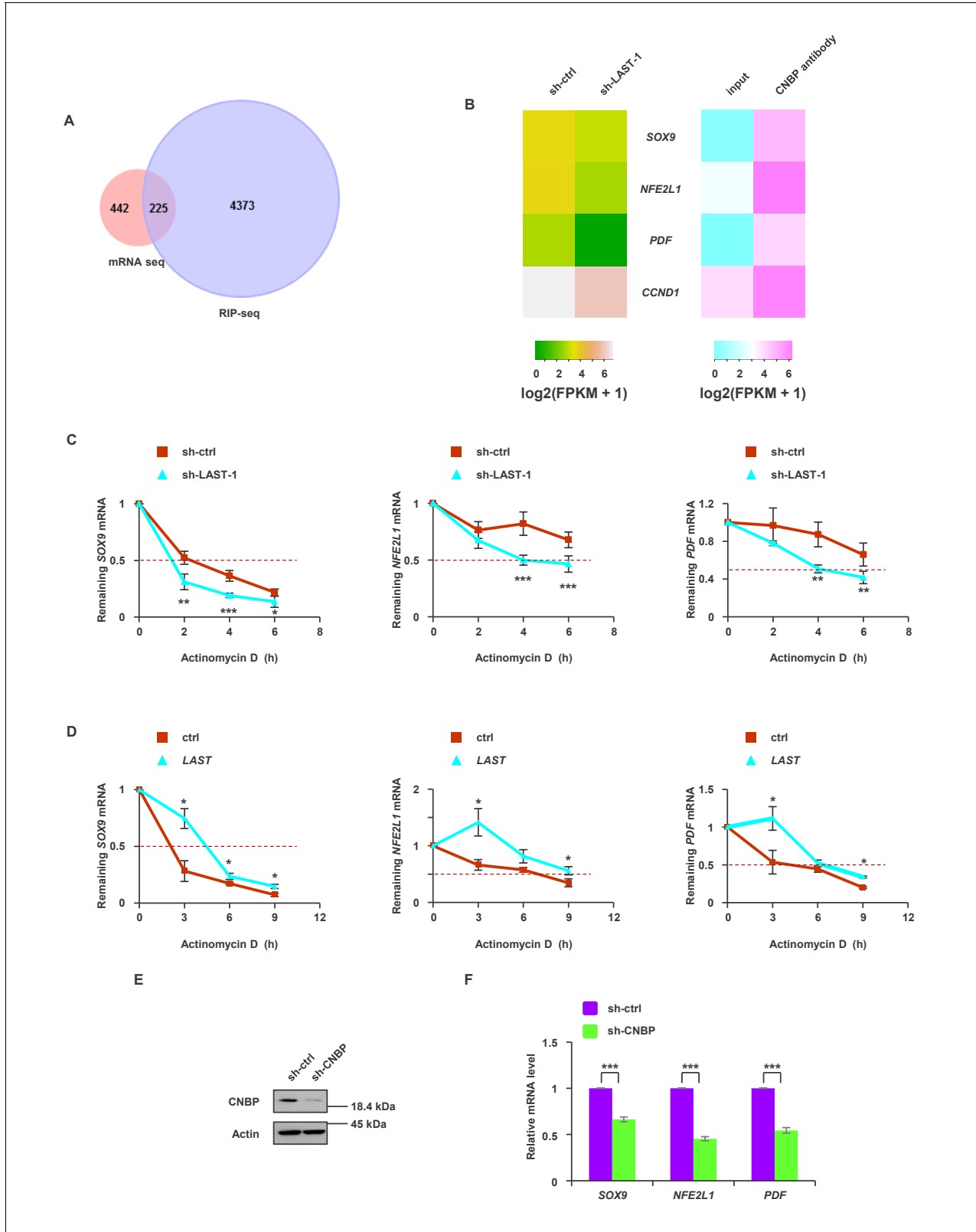

**Figure 5.** The synergistic effect of *LAST* and CNBP on mRNA expression. (**A**) Venn diagram represents 225 overlapping transcripts (*Supplementary file 2*) obtained from *LAST* RNA-seq (red) and CNBP RIP-seq (violet). (**B**) Heatmap showing that *SOX9*, *NFE2L1*, *PDF* and *CCND1* are not only decreased upon *LAST* knockdown but also enriched by CNBP. (**C**) HCT116 cells expressing control shRNA or *LAST* shRNA-1 were treated with actinomycin D (1 μg/mL) for the indicated periods of time. Total RNAs were extracted and then analyzed by real-time RT-PCR to examine the mRNA

*Figure 5 continued on next page*

*Figure 5 continued*

half-life of *SOX9*, *NFE2L1* and *PDF*. Data shown are the mean ± SD (n = 6; *p<0.05, **p<0.01, ***p<0.001, two-tailed t-test). (**D**) HCT116 cells expressing control RNA or *LAST* were treated with actinomycin D (1 μg/mL) for the indicated periods of time. Total RNAs were extracted and then analyzed by real-time RT-PCR to examine the mRNA half-life of *SOX9*, *NFE2L1* and *PDF*. (**E**) HCT116 cells were infected with lentiviruses expressing control shRNA or *CNBP* shRNA. Ninety-six hours later, cell lysates were subjected to western blotting to detect the CNBP knockdown efficiency. (**F**) HCT116 cells were infected with lentiviruses expressing control shRNA or *CNBP* shRNA. Ninety-six hours after injection, total RNA was analyzed by real-time RT-PCR with the indicated primers. Data shown are the mean ± SD (n = 3; ***p<0.001, two-tailed t-test).

DOI: https://doi.org/10.7554/eLife.30433.020

The following source data and figure supplement are available for figure 5:

**Source data 1.** Source data for *Figure 5C, D and F*.
DOI: https://doi.org/10.7554/eLife.30433.022
**Figure supplement 1.** Uncropped images of blots.
DOI: https://doi.org/10.7554/eLife.30433.021

## Discussion

Cyclin D1 is a critical regulator of CDK kinase, which regulates cell cycle progression at the G1 to S-phase transition. Pre- or mature *CCND1* mRNA is regulated at different hierarchical levels by multiple protein factors. Multiple classical transcriptional factors, such as c-Myc, E2F1, OCT1, RELA and c-Jun, have been reported to modulate *CCND1* at the transcriptional level (*Guo et al., 2011*). Epigenetic and post-transcriptional mechanisms are also involved in the regulation of *cyclin D1/CCND1* (16, 29, 30). Moreover, Pitx2 and HuR, which belong to the same ribonucleoprotein complex, also control the decay rate of *CCND1* mRNA (*Gherzi et al., 2010*). However, whether lncRNA(s) is (are) involved in the regulation of *CCND1* mRNA stability remains largely unaddressed. Very recently, NcRNA$_{CCND1}$ was reported to negatively regulate *CCND1* transcription by recruiting TLS to the *CCND1* promoter (*Wang et al., 2008*). In this study, we characterized an overlooked mechanism of *CCND1* mRNA regulation. c-Myc induced-*LAST* cooperates with CNBP, by which *LAST* is guided to the 5' untranslated region of *CCND1* messenger RNA and thus stabilizes *CCND1* mRNA (*Figure 6H*). The detailed mechanism underlying this 5'end protection requires further characterization.

Normal growth control depends on the architecture of precise cell cycle control, and disturbing any component of this network could result in neoplastic growth and tumorigenesis. The G1/S transition is a major checkpoint in cell cycle progression, as it is a 'point of no return' beyond which cells are committed to dividing. Cyclin D1, along with its catalytically active partner CDK4, is a positive cell cycle regulator that advances the cell cycle from G1 to S phase (*McKay et al., 2002*). Instead of protein factors, in this study, we found a novel long noncoding RNA, *LAST*, that ensures normal cell cycle progression. Lacking this lncRNA causes cell cycle arrest at the G1/S stage due to decreased cyclin D1 and attenuates tumor growth. Both the *LAST* and *CCND1* expression levels are higher in most tumor tissues than in their corresponding normal tissues (*Figure 6—figure supplement 1A*). Conceivably, *LAST* could be a potential target for new cancer therapeutics. However, a correlation between the expression levels of *CCND1* and *LAST* in the 15 tumor types examined was not found (*Supplementary file 4*). In addition, there was no difference in survival when tumors were divided into those expressing high versus low *CCND1* or *LAST*. These results imply that the regulation of *CCND1* is more complicated than we had anticipated, and new functions of *LAST* need to be characterized.

CNBP encodes a nucleic-acid binding protein that has seven zinc-finger domains and a preference for binding single-stranded DNA and RNA (*Flink and Morkin, 1995*). Previous studies have shown that CNBP acts on cap-independent translation of ornithine decarboxylase mRNA (*Sammons et al., 2010*) and also functions in sterol-mediated transcriptional regulation as well as c-Myc transcription (*Rajavashisth et al., 1989*; *Murphy et al., 2009*). In this study, we found that CNBP possesses a new function. CNBP is able to guide lncRNA to bind to the 5'UTR of *CCND1* mRNA, acting as a mediator between *LAST* and *CCND1* mRNA.

lncRNAs are able to regulate their genomic neighborhoods in cis (*Quinn and Chang, 2016*). Examples of cis-acting lncRNAs include enhancer RNAs (eRNAs) (*De Santa et al., 2010*), imprinted lncRNAs (*Mancini-Dinardo et al., 2006*; *Sleutels et al., 2002*) and dosage compensation lncRNAs

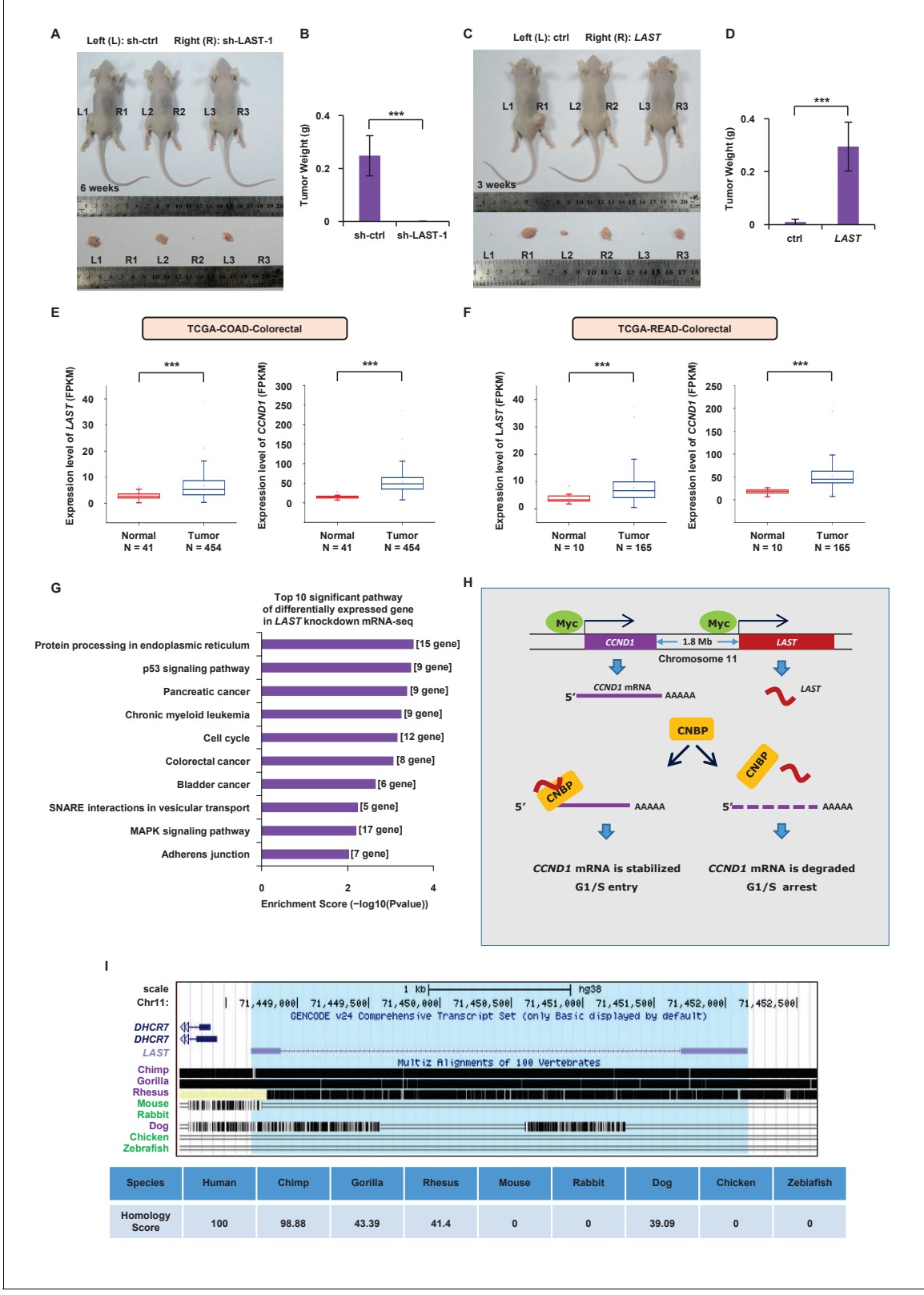

**Figure 6.** *LAST* promotes tumorigenesis. (**A**) A total of 3 × 10⁶ HCT116 cells expressing either control shRNA or *LAST* shRNA-1 were individually injected subcutaneously into the flanks of nude mice (n = 7 for each group) as indicated. Representative photographs of xenograft tumors in situ were taken 6 weeks after injection. (**B**) Tumors of the above nude mice (*Figure 6A*) were also selected to be weighed. Data shown are the mean ± SD (n = 7; ***p<0.001, two-tailed t-test). (**C**) A total of 3 × 10⁶ HCT116 cells expressing either control RNA or *LAST* were individually injected subcutaneously into

*Figure 6 continued*

the flanks of nude mice (n = 7 for each group) as indicated. Representative photographs of xenograft tumors in situ were taken 3 weeks after injection. (D) Tumors of the above nude mice (*Figure 6C*) were selected and weighed. Data shown are the mean ± SD (n = 7; ***p<0.001, two-tailed t-test). (E) Data for the *LAST* and *CCND1* expression levels in COAD (colon adenocarcinoma) tumor and normal tissues were downloaded from the TCGA dataset. Box plots showing differential expression of *LAST* and *CCND1* between normal (n = 41) and tumor (n = 454) samples. Statistical analysis was performed using the two-tailed t-test (***p<0.001). (F) Data for *LAST* and *CCND1* expression levels in READ (rectum adenocarcinoma) tumor and normal tissues were downloaded from the TCGA dataset. Box plots showing the differential expression of *LAST* and *CCND1* between normal (n = 10) and tumor (n = 165) samples. Statistical analysis was performed using the two-tailed t-test (***p<0.001). (G) Pathway analysis of differentially downregulated genes (log2 (fold change) below - 0.58 in RNA-seq) in HCT116 with and without *LAST* knockdown. The top 10 significant pathways with enrichment scores are shown. (H) A schematic illustration of the proposed model depicting the role of c-Myc-induced *LAST* in regulating *CCND1* mRNA stability via CNBP. (I) Gene homology analysis of *LAST* in human, chimp, gorilla, rhesus, mouse, rabbit, dog, chicken and zebrafish.

DOI: https://doi.org/10.7554/eLife.30433.023

The following source data and figure supplements are available for figure 6:

**Source data 1.** Source data for *Figure 6B, D, E and F*.

DOI: https://doi.org/10.7554/eLife.30433.026

**Figure supplement 1.** The expression levels of LAST and CCND1 are both higher in most tumor tissues than in their normal counterparts.

DOI: https://doi.org/10.7554/eLife.30433.024

**Figure supplement 2.** The expression levels of LAST and CCND1 are both higher in most tumor tissues than in their normal counterparts.

DOI: https://doi.org/10.7554/eLife.30433.025

(*Lee, 2012*; *Conrad and Akhtar, 2012*). Homo sapiens *cyclin D1/CCND1* and *LAST* are both located on chromosome 11, and the two genes are 1.8 Mb apart and in the same transcriptional direction (+strand). It is interesting to note that although *CCND1* and *LAST* are both subjected to positive transcriptional regulation by c-Myc, they do not share the same promoter. Rather, *CCND1* and *LAST* are transcribed separately by c-Myc via their respective promoters (*Figures 1K* and *6H*). The relatively long distance between the *CCND1* and *LAST* genes may preclude their direct interaction. Moreover, *LAST* was shown to borrow a trans-acting factor, CNBP, as a mediator to connect the 5'UTR of *CCND1* mRNA and itself, thus affecting *CCND1/cyclin D1* expression in trans. Without CNBP, *LAST* shows no effect on CCND1 (*Figure 3—figure supplement 1F*), further supporting the concept that *LAST* does not regulate CCND1 in cis. Therefore, co-location of *CCND1* and *LAST* on the same chromosome appears to be a random event.

In summary, our findings from this investigation have uncovered a novel, c-Myc-induced, long non-coding RNA, *LAST*. The *LAST* gene is encoded physically on the same chromosome as *CCND1*. Normally, *LAST* interacts with CNBP, a RNA binding protein, by which it is guided towards the 5'UTR of *CCND1* mRNA, leading to the stabilization of *CCND1* mRNA, which in turn ensures orderly cell cycle progression. In the case of *LAST* dysregulation, *CCND1* mRNA becomes unstable, resulting in decreased cyclin D1, inevitably causing cell cycle arrest and stoppage of cell division (*Figure 6H*). This is a novel mechanism for *CCND1* mRNA regulation. Based on the importance of cyclin D1 in proliferative control and its ability to promote oncogenic transformation, this finding provides new insight into the complexity of the regulatory network underlying the mechanistic regulation of *cyclin D1/CCND1*. Moreover, this *LAST*/CNBP regulatory mode can be applied to other genes; three different mRNAs, *SOX9*, *NFE2L1* and *PDF*, were identified with half-lives that were prolonged by *LAST*/CNBP. The lack of similarity between human *LAST* and transcripts of *Mus musculus* also precludes using mouse c-Myc-driven tumor models to further clarify the significance of the *LAST* in c-Myc-mediated cell cycle regulation and tumor growth in vivo (*Figure 6I*).

# Materials and methods

## Antibodies and reagents

The following antibodies were used for western blot analysis in this study: anti-c-Myc (Cell Signaling Technology); anti-GAPDH and anti-β-Actin (CMC-TAG); normal rabbit IgG, normal mouse IgG$_{2a}$, anti-HUR, anti-cyclin D1, anti-CNBP and anti-NADSYN1 (Santa Cruz); anti-FLAG (Sigma-Aldrich); anti-cyclin E1, anti-CDK2, and anti-CDK4 (ImmunoWay Biotechnology Company); anti-HNRNPK (ABclonal); anti-DHCR7 (ABCAM). Anti-c-Myc used for ChIP assay was from Santa Cruz. Thymidine, Nocodazole, Mimosine, EGF, hydrocortisone, Cholera Toxin, insulin and Doxycycline was from

Sigma-Aldrich. Actinomycin D was from Solarbio. Strepavidin beads for RNA pull-down assay was from Invitrogen.

## Cell culture

H1299, HCT116, IMR90, 293T and HAFF cell lines were cultured in DMEM (Dulbecco's modified Eagle's medium) medium containing 10% fetal bovine serum. P493-6 and MCF7 cell lines were cultured in RPMI medium 1640 containing 10% fetal bovine serum. MCF10A cell line were cultured in DMEM/F12 medium containing 5% horse serum, 20 µg/mL EGF, 0.5 µg/mL hydrocortisone, 100 ng/mL Cholera Toxin and 10 µg/mL insulin. P493-6 cells carrying a c-Myc tet-off system were provided by professor Ping Gao. All other cell lines were purchased from the American Type Culture Collection (ATCC, Manassas, VA, USA). All cells were tested by STR profiling (GenePrint 10 System kit from Promega and AuthentiFiler PCR Amplification Kit from ThermoFisher) to authenticate the identity. All cells were tested for mycoplasma contamination by Cell Culture Contamination Detection Kit (ThermoFisher).

## Western blotting, northern blotting and real-time RT-PCR

Western blotting, Northern blotting and real-time RT-PCR were performed as described previously (*Zhang et al., 2016*).

## Colony-formation assay

HCT116 cells ($1 \times 10^3$) expressing control shRNA, lncRNA-52 shRNA-1,–2, lncRNA-51 shRNA-1 or −2 were cultured in a six-well plate. Ten days later, cells were fixed, stained with crystal violet and photographed.

## Quantitationfor the expression levels of *LAST*

The exact copy numbers of *LAST* transcripts per HAFF, IMR90, MCF10A, HCT116, MCF7 or H1299 cell were quantified by using quantitative real-time RT-PCR assay. In this assay, serially diluted RT-PCR products of *LAST* were used as templates to formulate standard curves, and the exact copies of *LAST* per cell were calculated accordingly.

## ChIP assay

HCT116 cells were crosslinked with 1% formaldehyde for 10 min. The ChIP assay was performed by using anti-c-Myc antibody and the Pierce Agarose ChIP kit (ThermoScientific, USA) according to the manufacturer's instructions. Anti-Rabbit immunoglobulin G was used as a negative control. The bound DNA fragments were subjected to real-time PCR using the specific primers (*Table 1*).

## Luciferase reporter assay

To determine the effect of c-Myc on *LAST* promoter, either p3xflag-Myc-CMV-24 or p3xflag-Myc-CMV-24-c-Myc was co-transfected into HCT116 cells together with individual pGL3, pGL3-BS2, pGL3-BS2M, pGL3-BS3 or pGL3-BS3M construct plus Renilla luciferase reporter plasmid. Twenty-four hours after transfection, firefly and Renilla luciferase activity were measured by a Dual-Luciferase Reporter Assay System (Promega, Madison, WI, USA). The data are represented as mean ± SD of three independent experiments.

## Cell cycle analysis

HCT116 cells were infected with lentiviruses and screened by puromycin, followed by plating into 6 mm dishes. During the proliferative exponential phase (50% confluency), cells were fixed in 70% ethanol overnight. Cells were then stained with propidium iodide and analyzed by flow cytometry.

## RNA in situ hybridization

To detect *LAST*, RNA FISH was carried out as previously described with in vitro transcribed antisense probes labeled by Nucleic Acid Labeling Kits (Life technologies, USA) with Alexa Fluor 488 (*Yin et al., 2012*). The sequence of RNA probe was CGUCUUUUCAGGACACAAAGGCAUGCAGG UGCAUCAUCUCUCUCUAUUAACGGGUCAGCUGGUCGGCAUGGUCAGCUGGUCGGUGGUCUC UUAUUAGGAGAAAGUCACUGAAAUCAGUCUCUUGUCCAAUCACAGCUGCUAUGGCUGAUCG

**Table 1.** Oligomers used in this study

| Name | Application | Sequence |
| --- | --- | --- |
| qrt-lncRNA-5639-F | qRT-PCR | GACCTTGGGCTAGTTATTTTGTG |
| qrt-lncRNA-5639-R | qRT-PCR | TCCTCTCTCCTTTCCTGTCTG |
| qrt-lncRNA-51-F | qRT-PCR | ACCACAGATCCAGTAGCCTAG |
| qrt-lncRNA-51-R | qRT-PCR | CCTAACCACACTCCAAGACAC |
| qrt-lncRNA-5630-F | qRT-PCR | CTCCAACATCACCAAAACCAC |
| qrt-lncRNA-5630-R | qRT-PCR | TCTTGGCATGTGGTATCTGTC |
| qrt-lncRNA-5690-F | qRT-PCR | TCGACATGAAACTTGGGTGG |
| qrt-lncRNA-5690-R | qRT-PCR | GGCCAAATTCACTTGATGCTC |
| qrt-LAST-F | qRT-PCR | GGATCCTCCATAAACGATCAG |
| qrt-LAST-R | qRT-PCR | AGCTGGTCGGTGGTCTCTTA |
| qrt-CNBP-F | qRT-PCR | CCTCGGATAGAGGTTTCCAG |
| qrt-CNBP -R | qRT-PCR | ACCGCAGTTATAGCAGGCTT |
| qrt-CDK4-F | qRT-PCR | CTGGTGTTTGAGCATGTAGACC |
| qrt-CDK4-R | qRT-PCR | AAACTGGCGCATCAGATCCTT |
| qrt-CDK2-F | RT-PCR | GCTAGCAGACTTTGGACTAGCCAG |
| qrt-CDK2-R | qRT-PCR | AGCTCGGTACCACAGGGTCA |
| qrt-CCNB1-F | qRT-PCR | AAGAGCTTTAAACTTTGGTCTGGG |
| qrt-CCNB1-R | qRT-PCR | CTTTGTAAGTCCTTGATTTACCATG |
| qrt-CCNE1-F | qRT-PCR | ATCAGCACTTTCTTGAGCAACA |
| qrt-CCNE1-R | qRT-PCR | TTGTGCCAAGTAAAAGGTCTCC |
| qrt-CCND1-CDS-F | qRT-PCR | ACGAAGGTCTGCGCGTGTT |
| qrt-CCND1-CDS-R | qRT-PCR | CCGCTGGCCATGAACTACCT |
| qrt-CCND1-5′UTR-F | qRT-PCR | CTGGAGCCTCCAGAGGGCTGT |
| qrt-CCND1-5′UTR-R | qRT-PCR | GCGCTCCCTCGCGCTCTTC |
| qrt-CCND1-3′UTR-1-F | qRT-PCR | GGAAAGCTTCATTCTCCTTGTTG |
| qrt-CCND1-3′UTR-1-R | qRT-PCR | TTCTTTTGCTTAAGTCAGAGATGGAA |
| qrt-CCND1-3′UTR-2-F | RT-PCR | CATTGATTCAGCCTGTTTGG |
| qrt-CCND1-3′UTR-2-R | qRT-PCR | GAATTCATCGGAACCGAACT |
| qrt-CCND1-3′UTR-3-F | RT-PCR | TCTCAATGAAGCCAGCTCACA |
| qrt-CCND1-3′UTR-3-R | RT-PCR | TTTTGGTTCGGCAGCTTG |
| qrt-CCND1-intron-1-F | qRT-PCR | CTTTGTTCAAGCAGCGAGTC |
| qrt-CCND1-intron-1-R | qRT-PCR | AAGGTCCTCCAAGCCGATA |
| qrt-CCND1-intron-2-F | qRT-PCR | CCCAGCTCCCTTGAGTCC |
| qrt-CCND1-intron-2-R | qRT-PCR | CGGTCCTGGATGTTGGAG |
| qrt-CCND1-intron-3-F | qRT-PCR | TTTGTCATCGGCCAGAAATA |
| qrt-CCND1-intron-3-R | qRT-PCR | GACCTTCAGAGCACAGACCA |
| qrt-CCND1-intron-4-F | qRT-PCR | ATGTGCGTGGCCAATAAATA |
| qrt-CCND1-intron-4-R | qRT-PCR | ATCCCAGGGTTTAACAGCAG |
| qrt-c-Myc-F | qRT-PCR | AGCGACTCTGAGGAGGAAC |
| qrt-c-Myc-R | qRT-PCR | TGTGAGGAGGTTTGCTGTG |
| qrt-PDF-F | qRT-PCR | GCTGCGGCGCTCCTATT |
| qrt-PDF-R | qRT-PCR | TTGGCACACGTGCGAGAAC |
| qrt-NFE2L1-F | qRT-PCR | TGGCTATGGTATCCACCCCA |
| qrt-NFE2L1-R | qRT-PCR | ACCAGCCAGGCATTTACCTC |

*Table 1 continued on next page*

*Table 1 continued*

| Name | Application | Sequence |
| --- | --- | --- |
| qrt-SOX9-F | qRT-PCR | GCGAGCCCGATCTGAAGAAG |
| qrt-SOX9-R | qRT-PCR | GTTCTTGCTGGAGCCGTTGA |
| qrt-DHCR7-F | qRT-PCR | ATCTGCCATGACCACTTCGG |
| qrt-DHCR7-R | qRT-PCR | CAGACCCTGCAGCGTGTAAA |
| qrt-NADSYN1-F | qRT-PCR | GCCGTGAGGAGTGGAAATGA |
| qrt-NADSYN1-R | qRT-PCR | GTGGTCAGTATGCGTCCACA |
| qrt-TOMM6-F | qRT-PCR | TGCTGGCTCGGCTAATGAAA |
| qrt-TOMM6-R | qRT-PCR | TCCTATCAGTGGCAAAGCGG |
| qrt-CEBPG-F | qRT-PCR | GAGCATGCACACAACCTTGC |
| qrt-CEBPG-R | qRT-PCR | CATTGTCGCCATCTGCTGTC |
| qrt-PRNP-F | qRT-PCR | GGAGAACTTCACCGAGACCG |
| qrt-PRNP-R | qRT-PCR | AGGACCATGCTCGATCCTCT |
| qrt-CHMP1B-F | qRT-PCR | GTTCAACCTGAAGTTCGCGG |
| qrt-CHMP1B-R | qRT-PCR | GGCATTTTCGGCGTGTATCC |
| qrt-MSX1-F | qRT-PCR | CCACTCGGTGTCAAAGTGGA |
| qrt-MSX1-R | qRT-PCR | GAAGGGGACACTTTGGGCTT |
| qrt-THAP11-F | qRT-PCR | AACCTGGTATCTGCTTCCGC |
| qrt-THAP11-R | qRT-PCR | TGAGATCGATGGGCTTCACG |
| qrt-C16orf91-F | qRT-PCR | ATGGGAAAGGGACATCAGCG |
| qrt-C16orf91-R | qRT-PCR | CTCCCCACACCTGTCTCAAC |
| qrt-VMA21-F | qRT-PCR | CATCTGCACAGCACCTTACAGTTTGC |
| qrt-VMA21-R | qRT-PCR | GAAATGCAGCACATCCAAATCCTCCC |
| qrt-PLEC-F | qRT-PCR | CCGGGCAGTCTCTGAAGATG |
| qrt-PLEC-R | qRT-PCR | GCGTTTTCCCAAGGTTCCAG |
| qrt-DLG5-F | qRT-PCR | GATGACCCGGGAGAGAAACG |
| qrt-DLG5-R | qRT-PCR | GGATTCAGCCTGTGGTAGGG |
| qrt-EPPK1-F | qRT-PCR | GTGTGTGATGAGTGGCCACACC |
| qrt-EPPK1-R | qRT-PCR | CTCTGGGTACACTGGCCTGCTCT |
| qrt-HIST2H4A-F | qRT-PCR | GGCGGAAAAGGCTTAGGCAA |
| qrt-HIST2H4A-R | qRT-PCR | CCAGAGATCCGCTTAACGCC |
| qrt-MYH9-F | qRT-PCR | ATCTCGTGCTATCCGCCAAG |
| qrt-MYH9-R | qRT-PCR | GTTGTACGGCTCCAACAGGA |
| qrt-PPL-F | qRT-PCR | AGGCAAATACAGCCCCACTG |
| qrt-PPL-R | qRT-PCR | AGGTCACTCTGCATCTTGGC |
| qrt-PRKDC-F | qRT-PCR | GGACCTATAGCGTTGTGCCC |
| qrt-PRKDC-R | qRT-PCR | GATCACTCAGGTAAGCCGCC |
| qrt-GDF15-F | qRT-PCR | TCCAGATTCCGAGAGTTGCG |
| qrt-GDF15-R | qRT-PCR | CGAGGTCGGTGTTCGAATCT |
| qrt-Actin-F | qRT-PCR | GACCTGACTGACTACCTCATGAAGAT |
| qrt-Actin-R | qRT-PCR | GTCACACTTCATGATGGAGTTGAAGG |
| qrt-U6-F | qRT-PCR | GCTTCGGCAGCACATATACTAAAAT |
| qrt-U6-R | qRT-PCR | CGCTTCACGAATTTGCGTGTCAT |
| qrt-U1-F | qRT-PCR | GGCGAGGCTTATCCATTG |
| qrt-U1-R | qRT-PCR | CCCACTACCACAAATTATGC |

*Table 1 continued on next page*

*Table 1 continued*

| Name | Application | Sequence |
|------|-------------|----------|
| sh-LAST-F-1 | plasmid construction | ccggAAGAGGATCCTCCATAAACGActcgagTCGTTTATGGAGGATCCTCTTtttttg |
| sh-LAST-R-1 | plasmid construction | aattcaaaaaAAGAGGATCCTCCATAAACGActcgagTCGTTTATGGAGGATCCTCTT |
| sh-LAST-F-2 | plasmid construction | ccggTCAGCCATAGCAGCTGTGATTctcgagAATCACAGCTGCTATGGCTGAtttttg |
| sh-LAST-R-2 | plasmid construction | aattcaaaaaTCAGCCATAGCAGCTGTGATTctcgagAATCACAGCTGCTATGGCTGA |
| sh-lncRNA-51-F-1 | plasmid construction | ccggAAGCAGATGGAGGGAAGTTggatcc AACTTCCCTCCATCTGCTTtttttg |
| sh-lncRNA-51-R-1 | plasmid construction | aattcaaaaaAAGCAGATGGAGGGAAGTTggatccAACTTCCCTCCATCTGCTT |
| sh-lncRNA-51-F-2 | plasmid construction | ccggGGAAGCAGAGTAAGCAAGTGAGGATCCTCACTTGCTTACTCTGCTTCCtttttg |
| sh-lncRNA-51-R-2 | plasmid construction | aattcaaaaaGGAAGCAGAGTAAGCAAGTGAGGATCCTCACTTGCTTACTCTGCTTCC |
| LAST-DNA-1-sense | lncRNA pull down | (biotin-)TAAACGATCAGCCATAGCA |
| LAST-DNA-1-antisense | lncRNA pull down | (biotin-)TGCTATGGCTGATCGTTTA |
| LAST-DNA-2-sense | lncRNA pull down | (biotin-)TCATCGTGCCTCAGTTTCC |
| LAST-DNA-2-antisense | lncRNA pull down | (biotin-)GGAAACTGAGGCACGATGA |
| LAST-DNA-3-sense | lncRNA pull down | (biotin-)ACAGACACAGTTCTTGGTC |
| LAST-DNA-3-antisense | lncRNA pull down | (biotin-)GACCAAGAACTGTGTCTGT |
| LAST-DNA-4-sense | lncRNA pull down | (biotin-)ATGGGTCATATATTACATG |
| LAST-DNA-4-antisense | lncRNA pull down | (biotin-)CATGTAATATATGACCCAT |
| LAST-DNA-5-sense | lncRNA pull down | (biotin-)GTTGAATATGTATGTTTAG |
| LAST-DNA-5-antisense | lncRNA pull down | (biotin-)CTAAACATACATATTCAAC |
| LAST-DNA-6-sense | lncRNA pull down | (biotin-)CCAGCCTCAGACAGATGGC |
| LAST-DNA-6-antisense | lncRNA pull down | (biotin-)GCCATCTGTCTGAGGCTGG |
| CCND1-DNA-1-sense | mRNA pull down | (biotin-)GCGCAGTAGCAGCGAGCAGCA |
| CCND1-DNA-1-antisense | mRNA pull down | (biotin-)TGCTGCTCGCTGCTACTGCGC |
| CCND1-DNA-2-sense | mRNA pull down | (biotin-)CCGCTGGCCATGAACTACCTG |
| CCND1-DNA-2-antisense | mRNA pull down | (biotin-)CAGGTAGTTCATGGCCAGCGG |
| CCND1-DNA-3-sense | mRNA pull down | (biotin-)AACACGCGCAGACCTTCGTTG |
| CCND1-DNA-3-antisense | mRNA pull down | (biotin-)CAACGAAGGTCTGCGCGTGTT |
| CCND1-DNA-4-sense | mRNA pull down | (biotin-)CGTAGGTAGATGTGTAACCTCT |
| CCND1-DNA-4-antisense | mRNA pull down | (biotin-)AGAGGTTACACATCTACCTACG |
| CCND1-DNA-5-sense | mRNA pull down | (biotin-)AGAGTCATCTGATTGGACAGGC |
| CCND1-DNA-5-antisense | mRNA pull down | (biotin-GCCTGTCCAATCAGATGACTCT |
| CCND1-DNA-6-sense | mRNA pull down | (biotin-)AATGAAGCCAGCTCACAGTGCT |
| CCND1-DNA-6-antisense | mRNA pull down | (biotin-)AGCACTGTGAGCTGGCTTCATT |
| ChIP-LAST-a-F | qRT-PCR for ChIP | TTCCTGACAGCAGATTCCAG |
| ChIP-LAST-a-R | qRT-PCR for ChIP | TCTGCCATGTTTGGAGAATG |
| ChIP-LAST-b-F | qRT-PCR for ChIP | ACCTGCTCACCTGGGCAAGC |
| ChIP-LAST-b-R | qRT-PCR for ChIP | GGCAATCGCTGACATCATCCGGG |
| ChIP-LAST-c-F | qRT-PCR for ChIP | GGGATCCCAGCTGACCAGCTG |
| ChIP-LAST-c-R | qRT-PCR for ChIP | GAGGCACGATGATCCAGGTGATGAG |
| ChIP-LAST-d-F | qRT-PCR for ChIP | CTGAGCCACAGTGCGAGCCG |
| ChIP-LAST-d-R | qRT-PCR for ChIP | GACAGTAAGGCCTGTTACCCGAGC |
| ChIP-LAST-e-F | qRT-PCR for ChIP | AAGTCAAACAGCACGAACCC |
| ChIP-LAST-e-R | qRT-PCR for ChIP | CGGATGGGCATTGACGTTAT |
| ChIP-LAST-f-F | qRT-PCR for ChIP | TCAAGTGCAGTTCCTGTAGTTTC |
| ChIP-LAST-f-R | qRT-PCR for ChIP | GATGGCGCTGAATTCTTGGGAACC |

*Table 1 continued on next page*

*Table 1 continued*

| Name | Application | Sequence |
| --- | --- | --- |
| ChIP-LAST-g-F | qRT-PCR for ChIP | TCCCTTCTTGTCCCTTCAAA |
| ChIP-LAST-g-R | qRT-PCR for ChIP | CCTAAAGACCAACGGGAAAC |
| ChIP-LAST-h-F | qRT-PCR for ChIP | TCTAGGGTTCTGGGCTGTCT |
| ChIP-LAST-h-R | qRT-PCR for ChIP | GTCAGGCTCACGAGACGAT |

DOI: https://doi.org/10.7554/eLife.30433.027

UUUAUGGAGGAUCCUCUUCGCCCCGGGACGUGAGCCCUAGGACCAAGAACUGUGUCUG UUUUGCUCCUUGCGGUGCACCGGCGCCUGGACAUACGCUCCAUCAAUGUGCGUCGC-GAGCCGCUGAAGCCCCAUUUGCCGAGGGGGAAACUGAGGCACGAUG. The nuclei were counterstained with PI.

## Cytosolic/nuclear fractionation

HCT116 cells ($1 \times 10^7$) were incubated with hypotonic buffer (25 mM Tris-HCl, PH 7.4, 1 mM $MgCl_2$, 5 mM KCl) on ice for 5 min. An equal volume of hypotonic buffer containing 1% NP-40 was then added, and each sample was left on ice for another 5 min. After centrifugation at 5000 g for 5 min, the supernatant was collected as the cytosolic fraction. The pellets were re-suspended in nucleus resuspension buffer (20 mM HEPES, PH 7.9, 400 mM $NaCl_2$, 1 mM EDTA, 1 mM EGTA, 1 mM DTT, 1 mM PMSF), and incubated at 4℃ for 30 min. Nuclear fraction was collected after removing insoluble membrane debris by centrifugation at 12000 g for 10 min.

## RNA immunoprecipitation RT-PCR

RNA immunoprecipitation (RIP) was performed as described previously (*Yang et al., 2014*). $1 \times 10^7$ cells were lysed in RIP buffer supplemented with RNase A inhibitor and DNase I before centrifugation. Cell lysates were precleared with protein A/G beads (Pierce) before they were incubated with protein A/G beads coated with the indicated antibodies at 4℃ for 3 hr. After extensive washing, the bead-bound immunocomplexes were eluted using elution buffer (50 mM Tris [pH 8.0], 1% SDS, and 10 mM EDTA) at 65℃ for 10 min. To isolate protein-associated RNAs from the eluted immunocomplexes, samples were treated with proteinase K, and RNAs were extracted by phenol/chloroform. Purified RNAs were then subjected to RT-PCR analysis.

## RIP-seq (RIP sequencing)

RIP was performed as described previously (*Xiang et al., 2014*). Briefly, two 10 $cm^2$ dishes of HCT116 cells were washed three times with cold PBS and irradiated at 200 mJ/$cm^2$ at 254 nm in HL-2000 HybriLinker™ UV Crosslinker. Cells were collected and resuspended in 1 ml RIP buffer. Cells were then homogenized and followed by 3 rounds of sonication on ice. Cell lysates were precleared with protein A/G beads (Pierce) before they were incubated with protein A/G beads coated with the indicated antibodies at 4℃ for 3 hr. After extensive washing, the bead-bound immunocomplexes were eluted using elution buffer (50 mM Tris [pH 8.0], 1% SDS, and 10 mM EDTA) at 65℃ for 10 min. To isolate protein-associated RNAs from the eluted immunocomplexes, samples were treated with proteinase K, and RNAs were extracted by phenol/chloroform. The sequencing was performed and analyzed by KangChen Bio-tech, Shanghai, China. The sequencing data were deposited in the National Center for Biotechnology Information Gene Expression Omnibus database (GSE106918).

## mRNA-seq (mRNA sequencing)

Total RNA from HCT116 cells expressing either control shRNA or *LAST* shRNA-1 was extracted by phenol/chloroform. The mRNA-seq was performed and analyzed by KangChen Bio-tech, Shanghai, China. The sequencing data were deposited in the National Center for Biotechnology Information Gene Expression Omnibus database (GSE106917).

## G-rich motifs analysis in RIP sequencing data

The peak-calling tool MACS2 (*Zhang et al., 2008*) (https://github.com/taoliu/MACS/) with default parameter settings was used to call enriched peaks with *RIP.bed* as input and *input.bed* as control. A PERL script was written to calculate the proportion of the peaks containing each of the five given motifs (TGGAGNW, TGGAG, GGAGNW, GGAG and GGR) in all the RIP peaks. To test the significance of G-rich motif enrichment, another PERL script was used to perform statistical simulations by generating 1000 random samples of DNA sequences with the same size and the same length distribution as that of the RIP peaks. For each given motif, the average proportion (with standard deviation) of motif-containing sequences in random samples was calculated. A U-test was performed for each G-rich motif to test the significance of the difference between the proportion of the motif-containing sequences in RIP peaks and that in random DNA samples.

## Biotin pull-down assay

All processes were performed in the RNase-free conditions. For antisense oligomer affinity pull-down assay, sense or antisense biotin-labeled DNA oligomers corresponding to *LAST* or *CCND1* mRNA (1 μM) were incubated with lysates from HCT116 cells ($1 \times 10^7$) or the cytosolic/nuclear extracts. One hour after incubation, streptavidin-coupled agarose beads (Invitrogen) were added to isolate the RNA-protein complex or RNA-RNA complex. For in vitro RNA pull-down assay, 5 μg in vitro-synthesized biotin-labeled RNA was incubated with lysates from HCT116 cells ($1 \times 10^7$) for 3 hr. Streptavidin-coupled agarose beads (Invitrogen) were then added to the reaction mix to isolate the RNA-protein complex or RNA-RNA complex. Immunocomplexes were then analyzed by real-time RT-PCR or western blotting.

## Electrophoretic mobility shift assay

The electrophoretic mobility shift assay (EMSA) was performed by using an EMSA/gel shift kit (Beyotime, China). Flag-CNBP protein was purified from 293T cells expressing Flag-CNBP. The biotin-labeled RNA fragments (as shown in *Figure 4D and E*) in vitro transcribed by T7 Transcription Kit (Epicentre, USA) were used in EMSA.

## Xenograft mouse model

HCT116 cells expressing control RNA or *LAST* ($3 \times 10^6$) were subcutaneously injected into the dorsal flank of 4-week-old male athymic nude mice (Shanghai SLAC Laboratory Animal Co. Ltd.) (n = 7 mice per group). After 3 weeks, mice were sacrificed, and tumors were excised and weighed. HCT116 cells expressing control shRNA or *LAST* shRNA-1 ($3 \times 10^6$) were subcutaneously injected into the dorsal flank of 4-week-old male athymic nude mice (Shanghai SLAC Laboratory Animal Co. Ltd.) (n = 7 mice per group). After 6 weeks, mice were sacrificed, and tumors were excised and weighed. Mice were randomly assigned to different experimental groups. During testing the tumors' weight, the experimentalists were blinded to the information and shape of tumor tissue masses. Studies on animals were conducted with approval from the Animal Research Ethics Committee of the University of Science and Technology of China (Permit Number: USTCACUC1701003).

## Acknowledgements

The results shown in *Figure 6E and F*, *Figure 6—figure supplements 1* and *2* are based upon data generated by the TCGA Research Network: http://cancergenome.nih.gov/. We would like to thank Professor Ping Gao for providing P493-6 cells carrying a c-Myc tet-off system. This work was supported by grants from the National Key R and D Program of China (2016YFC1302302) and National Natural Science Foundation of China (81430065 and 31371388).

## Additional information

### Funding

| Funder | Grant reference number | Author |
|---|---|---|
| Ministry of Science and Technology of the People's Republic of China | 2016YFC1302302 | Mian Wu |
| National Natural Science Foundation of China | 81430065 | Mian Wu |
| National Natural Science Foundation of China | 31371388 | Mian Wu |

The funders had no role in study design, data collection and interpretation, or the decision to submit the work for publication.

### Author contributions

Limian Cao, Conceptualization, Data curation, Formal analysis, Validation, Investigation, Visualization, Methodology, Writing—original draft, Writing—review and editing; Pengfei Zhang, Conceptualization, Data curation, Formal analysis, Validation, Investigation, Visualization, Writing—original draft, Writing—review and editing; Jinming Li, Formal analysis, Visualization, Methodology; Mian Wu, Conceptualization, Resources, Supervision, Funding acquisition, Writing—original draft, Project administration, Writing—review and editing

### Author ORCIDs

Mian Wu (iD) http://orcid.org/0000-0002-2714-0500

### Ethics

Animal experimentation: Studies on animals in this paper were conducted with approval from the Animal Research Ethics Committee of the University of Science and Technology of China (Permit Number: USTCACUC1701003).

### Decision letter and Author response

Decision letter https://doi.org/10.7554/eLife.30433.049
Author response https://doi.org/10.7554/eLife.30433.050

## Additional files

### Supplementary files

• Supplementary file 1. lncRNA expression microarray data. Information of the selected lncRNAs is colored.
DOI: https://doi.org/10.7554/eLife.30433.028

• Supplementary file 2. Overlap of the CNBP RIP sequencing dataset and *LAST* knockdown mRNA sequencing dataset (downregulation). Information of *SOX9*, *PDF*, *NFE2L1* and *CCND1* is colored.
DOI: https://doi.org/10.7554/eLife.30433.029

• Supplementary file 3. *LAST* knockdown mRNA sequencing dataset (downregulation).
DOI: https://doi.org/10.7554/eLife.30433.030

• Supplementary file 4. Correlation between *LAST* and *CCND1* expression levels in fifteen TCGA tumor types.
DOI: https://doi.org/10.7554/eLife.30433.031

• Transparent reporting form
DOI: https://doi.org/10.7554/eLife.30433.032

### Major datasets

The following datasets were generated:

| Author(s) | Year | Dataset title | Dataset URL | Database, license, and accessibility information |
|---|---|---|---|---|
| Wu M, Zhang P, Cao L | 2017 | Myc regulates gene expression in P493-6 cell lines which carry a Myc tet-off system | https://www.ncbi.nlm.nih.gov/geo/query/acc.cgi?acc=GSE106916 | Publicly available at the NCBI Gene Expression Omnibus (accession no: GSE10 6916) |
| Wu M, Cao L, Zhang P | 2017 | The impact of lncRNA LAST knockdown on gene expression profile in HCT116 cells | https://www.ncbi.nlm.nih.gov/geo/query/acc.cgi?acc=GSE106917 | Publicly available at the NCBI Gene Expression Omnibus (accession no: GSE10 6917) |
| Wu M, Cao L, Zhang P | 2017 | Identification of CNBP binding mRNA and the binding sites/motifs in HCT116 cells | https://www.ncbi.nlm.nih.gov/geo/query/acc.cgi?acc=GSE106918 | Publicly available at the NCBI Gene Expression Omnibus (accession no: GSE10 6918) |

The following previously published datasets were used:

| Author(s) | Year | Dataset title | Dataset URL | Database, license, and accessibility information |
|---|---|---|---|---|
| The Cancer Genome Atlas Research Network, Weinstein JN, Collisson EA, Mills GB, Mills Shaw KR, Ozenberger BA, Ellrott K, Shmulevich I, Sander C, Stuart JM | 2013 | The Cancer Genome Atlas | https://cancergenome.nih.gov/ | The National Cancer Institute (NCI) provides access to all individuals seeking information on www.cancer.gov, including individuals who are disabled. To provide this information, the NCI website complies with Section 508 of the Rehabilitation Act (as amended). |
| Mathelier A, Fornes O, Arenillas DJ, Chen C, Denay G, Lees J, Shai W, Shyr C, Tan G, Worsley-Hunt R, Zhang AW, Parcy F, Lenhard B, Sandelin A, Wasserman WW | 2016 | The high-quality transcription factor binding profile database (JASPAR) | http://jaspar.genereg.net/ | The database is ready to be deployed quickly for genome-wide studies through the JASPAR API. |

| Finn RD, Attwood TK, Babbitt PC, Bateman A, Bork P, Bridge AJ, Chang H, Dosztanyi Z, El-Gebali S, Fraser M, Gough J, Haft D, Holliday GL, Huang H, Huang X, Letunic I, Lopez R, Lu S, Marchler-Bauer A, Mi H, Mistry J, Natale DA, Necci M, Nuka G, Orengo CA, Park Y, Pesseat S, Piovesan D, Potter SC, Rawlings ND, Redaschi N, Richardson L, Rivoire C, Sangrador-Vegas A, Sigrist C, Sillitoe I, Smithers B, Squizzato S, Sutton G, Thanki N, Thomas PD, Tosatto SCE, Wu CH, Xenarios I, Yeh L, Young S, Mitchell AL | 2009 | InterPro: protein sequence analysis & classification | http://www.ebi.ac.uk/interpro/ | New SOAP-based Web Services have been added to complement the existing InterProScan Web Service. These allow users to programmatically retrieve InterPro entry data such as the abstract, integrated signature lists or GO terms. Users can download a range of clients from http://www.ebi.ac.uk/Tools/webservices/clients/dbfetch, including PERL, C#.NET and Java clients, to access this data. |
| ENCODE Project Consortium | 2004 | Encyclopedia of DNA Elements | https://www.encodeproject.org/ | Publicly available at ENCODE |

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
