## [Decision Letter]

Thank you for submitting your article "LncRNA-CCND1, a c-Myc-inducible long noncoding RNA, cooperates with CNBP to promote CCND1 mRNA stability" for consideration by *eLife*. Your article has been reviewed by three peer reviewers, one of whom is a member of our Board of Reviewing Editors and the evaluation has been overseen by Naama Barkai as the Senior Editor. The reviewers have opted to remain anonymous.

The reviewers have discussed the reviews with one another and the Reviewing Editor has drafted this decision to help you prepare a revised submission.

Summary:

In this manuscript Cao and colleagues describe the function of a lncRNA that they name lncRNA-CCDN1. It is transcriptionally regulated by c-Myc and necessary for cell cycle progression. The authors found that lncRNA-CCDN1 promotes the stability of CCDN1 mRNA by binding to it through the protein CNBP. They finally show that lncRNA-CCDN1 is upregulated in some cancers, and its downregulation inhibits tumor growth in xenografts.

The authors make a quite convincing case for the claim that lncRNA-CCND1 post-transcriptionally regulates the accumulation in the cytoplasm of the CCND1 RNA. The experiments are performed overall to a high standard, and since there are few convincing reports of lncRNAs regulating mRNA stability in the cytoplasm, this study is of high interest to the lncRNA community and the broader audience interested in gene regulation. However, some aspects need clarification and improvement prior to decision on publication.

Essential revisions:

1) The name of the lncRNA is not appropriate. Since the authors show that exogenous over-expression of lncRNA-CCND1 is sufficient for regulation of CCND1, and a rescue experiment with shRNA-insensitive lncRNA-CCND1, this lncRNA appears to act in trans. Therefore, the localization of lncRNA-CCND1 and CCND1 on the same chromosome is inconsequential, and since there is no evidence for direct pairing between the two RNA molecules, it’s also not clear that CCND1 is the main target of lncRNA-CCND1. There is no justification for the lncRNA-CCND1 name and the authors should choose a more appropriate one.

2) The authors should check the effect of the shRNA knockdown of lncRNA-CCND1 on the two adjacent genes DHCR7 and NADSYN1.

3) Because the sequence of the lncRNA is composed mostly of transposable elements, there is a concern about the specificity of the FISH probe, which does not appear to be listed in the supplementary materials. The authors should provide the sequence and show specificity of the signal using their shRNA KD cells, and to also show that the RNA is cytoplasmic in their subcellular fractionation data.

4) The TCGA analysis is lacking – its currently showing data from just two tumor types. They authors should show a comparison of normal and tumor expression levels for lncRNA-CCND1 in all the solid tumors in which TCGA has data (~15 tumor types), and show the same comparison for CCND1 levels, as well as the correlation between CCND1, MYC and lncRNA-CCND1 expression levels in each tumor type. If tumors are divided into those expressing high versus low cyclin D1 or lncRNA-CCND1, is there a difference in survival (Kaplan Meier plots).

5) The authors show some CLIP-seq data in Figure 5, but it’s not described at all and not used in the rest of the study. They should describe in detail how these data were obtained and what is their quality. Do they see CLIP peaks on CCND1 and lncRNA-CCND1? Do they see G-rich motifs in the CLIP peaks when considering the whole transcriptome? If the data is not of proper quality, it is best to leave it out of the manuscript.

6) The analysis of the transcripts that are both bound by CNBP and changed upon lincRNA KD should be part of the results and not the discussion. This part of the analysis is incomplete and the conclusions are not fully supported by the data. Is the overlap significantly higher than expected by chance? How many of the 10 mRNAs have different half-lifes following lincRNA-CCND1 KD and overexpression? Without these data the claim that this lincRNA can regulate other transcripts in the same way is not supported.

[Editors' note: further revisions were requested prior to acceptance, as described below.]

Thank you for resubmitting your work entitled "LAST, a c-Myc-inducible long noncoding RNA, cooperates with CNBP to promote CCND1 mRNA stability in human cells" for further consideration at *eLife*. Your revised article has been favorably evaluated by Naama Barkai (Senior editor), a Reviewing editor, and two reviewers.

The manuscript has been improved but there are some remaining issues that need to be addressed before acceptance, as outlined below:

Major comments:

1) It is not clear how the RIP-seq experiment was done – the authors refer to Yang et al., (2014), but it has RIP qRT-PCR and not RIP-seq. It is not clear if any digestion of the RNA following IP is performed. If there is no digestion – how can the authors conclude from it where CNBP is binding the CCND1 mRNA? If there is no digestion, then the peaks merely correspond to regions that are resistant to degradation, and it’s not surprising that they G/C-rich.

2) The manuscript still needs very extensive language editing. Sentences like "In addition, c-Myc has been reported to repress transcription of CCND1 mediated by core elements of the CCND1 promoter but not MAX when expression of human Myc in BALB/c-3T3 cells, Rat6 cells and rat embryo fibroblasts." (from the Introduction) are not comprehendible. Other examples: "was particularly attracted our attention", "G1-relevant cyclins and CDKs genes", "To further confirm that LAST affect CCND1 mRNA stability is through CNBP", "It is therefore expected that knockdown of CNBP will reduced the association between LAST and CCND1 mRNA", "The exact reason for LAST or 5-'UTR involved in choosing the CNBP domain(s) for association is not clear at this moment", "Three mRNAs namely *SOX9, NFE2L1* and *PDF* as the same as CCND1", "Six weeks after cells expressing shRNA injection or three weeks after cells expressing lncRNA injection, Due to the humane care for experimental animals,". These are just some of the examples. In present form, the article cannot be published in *eLife* without editing.

3) The lack of any anti-correlation between CCND1 and LAST across tumor samples, which should be discussed in the manuscript. It’s negative data, but important negative data, and so cannot be left out.

4) We cannot find the accession number for the mRNA-seq, microarray and RIP-seq results in the manuscript. Was the raw data deposited?

---

## [Author Response]

Essential revisions:1) The name of the lncRNA is not appropriate. Since the authors show that exogenous over-expression of lncRNA-CCND1 is sufficient for regulation of CCND1, and a rescue experiment with shRNA-insensitive lncRNA-CCND1, this lncRNA appears to act in trans. Therefore, the localization of lncRNA-CCND1 and CCND1 on the same chromosome is inconsequential, and since there is no evidence for direct pairing between the two RNA molecules, it’s also not clear that CCND1 is the main target of lncRNA-CCND1. There is no justification for the lncRNA-CCND1 name and the authors should choose a more appropriate one.

We agree that name of this lncRNA in previous manuscript is not very appropriate. Besides CCND1 mRNA, we showed in this revised manuscript that this lncRNA is able to stabilize at least three additional mRNA transcripts, including *SOX9, NFE2L1* and *PDF* (Figure 5). We therefore renamed this lncRNA as LAST (LncRNA-Assisted Stabilization of Transcript). Literally, it means this lncRNA makes transcript last longer. Accordingly, sentences have been added in text to state this (subsection “Identification of LAST, a c-Myc responsive long noncoding RNA that promotes cell proliferation”).

2) The authors should check the effect of the shRNA knockdown of lncRNA-CCND1 on the two adjacent genes DHCR7 and NADSYN1.

We have checked the effect of LAST knockdown on its two adjacent genes DHCR7 and NADSYN1, and found that LAST showed no effect on either mRNA or protein level of those two genes (Figure 2—figure supplement 1). This also implies that the decrease inCCND1 mRNA by knockdown of LAST is not an off-target effect. We have added sentences in text (subsection “LAST promotes G1/S transition and upregulates cyclin D1/CCND1”).

3) Because the sequence of the lncRNA is composed mostly of transposable elements, there is a concern about the specificity of the FISH probe, which does not appear to be listed in the supplementary materials. The authors should provide the sequence and show specificity of the signal using their shRNA KD cells, and to also show that the RNA is cytoplasmic in their subcellular fractionation data.

According to the reviewers’ suggestion, we have listed the FISH probe against LAST in the Materials and methods section (RNA in situ hybridization). Additionally, we performed a FISH experiment in H1299 cells expressing control shRNA, LAST shRNA-1 or -2 and found that the LAST signal intensity was markedly reduced in LAST depleted cells (Figure 1—figure supplement 1). These data showed that this probe we used is specific for LAST. Furthermore, the cytosolic localization of LAST was also confirmed by a new sub-cellular fractionation experiment shown in Figure 1. Sentences have been added in text to state this (subsection “Identification of LAST, a c-Myc responsive long noncoding RNA that promotes cell proliferation”).

4) The TCGA analysis is lacking – its currently showing data from just two tumor types. They authors should show a comparison of normal and tumor expression levels for lncRNA-CCND1 in all the solid tumors in which TCGA has data (~15 tumor types), and show the same comparison for CCND1 levels, as well as the correlation between CCND1, MYC and lncRNA-CCND1 expression levels in each tumor type. If tumors are divided into those expressing high versus low cyclin D1 or lncRNA-CCND1, is there a difference in survival (Kaplan Meier plots).

We have analyzed the LAST and CCND1expressionlevels in 15 types of normal and tumor tissues from TCGA. Both LAST and CCND1 expression levels were higher in the majority of tumor tissues relative to normal tissues (Figure 6, Figure 6—figure supplement 1 and Figure 6—figure supplement 2). Sentences have been added in text to state this (subsection “LASTpromotes tumorigenesis”). However, the correlation between CCND1, Myc and LAST expression levels in 15 types of tumor examined was not found (Data are shown in below Table R1). In addition, there is no difference in survival when tumors are divided into those expressing high versus low cyclin D1 or LAST. As these results do not add significantly to the main message, we have not included them in the revised manuscript.

TCGA Tumor TypesPearson Correlation Coefficient (LAST-CCND1)P valuePearson Correlation Coefficient (LAST-Myc)P valuePearson Correlation Coefficient (CCND1- Myc)P valueSample SizeBLCA0.0280.567-0.1050.034-0.080.107408BRCA0.11900.1180-0.0690.0231090CESC0.0890.122-0.0050.93-0.0440.445304CHOL-0.0880.610.0250.884-0.1870.27436COAD-0.1620.0010.18800.0730.119454ESCA0.3370-0.1320.0940.0210.788161HNSC0.0340.442-0.0680.131-0.0310.496499KIRC0.0080.8450.0370.3990.0910.037530LIHC-0.0470.366-0.0140.7830.0030.949371LUAD-0.0370.402-0.0610.169-0.0330.454513PAAD-0.0250.7450.0220.7720.1540.041177PRAD-0.0610.174-0.0480.29-0.0160.715495READ-0.27700.0720.3570.0630.422165STAD0.1670.0010.0730.1590.0450.384375UCEC0.0240.58-0.0240.584-0.0490.25543

BLCA (bladder urothelial carcinoma); BRCA (breast invasive carcinoma); CESC (cervical squamous cell carcinoma); CHOL (cholangiocarcinoma); COAD (colon adenocarcinoma); ESCA (esophageal carcinoma); HNSC (head and neck squamous carcinoma); KIRC (kidney renal clear cell carcinoma); LIHC (liver hepatocellular carcinoma); LUAD (lung adenocarcinoma); PAAD (pancreatic adenocarcinoma); PRAD (prostate adenocarcinoma); READ (rectum adenocarcinoma); STAD (stomach adenocarcinoma); UCEC (uterine corpus endometrial carcinoma).

5) The authors show some CLIP-seq data in Figure 5, but it’s not described at all and not used in the rest of the study. They should describe in detail how these data were obtained and what is their quality. Do they see CLIP peaks on CCND1 and lncRNA-CCND1? Do they see G-rich motifs in the CLIP peaks when considering the whole transcriptome? If the data is not of proper quality, it is best to leave it out of the manuscript.

We appreciate very much for referees’ comment. We have analyzed the CNBP RIP-seq enrichment peaks and a *prominent* peak on CCND1 5’UTR was observed (Figure 4—figure supplement 1). This is consistent with the conclusion from Figure 4. Because the sequencing depth of CNBP RIP-seq is 6 G, it is not enough to analyze the peaks on noncoding RNA. However, we have evaluated the enrichment efficiency of LAST and CCND1 in the RIP-seq samples. Both LAST and CCND1 were significantly enriched by CNBP antibody (Author response image 1). Furthermore, we checked the proportion of G-rich motifs in all peak sequences from CNBP RIP samples and found that nearly 60% of the CNBP enriched sequences contained GGAG core (Figure 4—figure supplement 2), which is in good agreement with what had been previously reported (1, 2). These results prove that CNBP RIP-seq data are reliable. Sentences have been added in text to indicate these results (subsection “Both LAST and CCND1 mRNA bind to CNBP through their G-rich motifs”).

6) The analysis of the transcripts that are both bound by CNBP and changed upon lincRNA KD should be part of the results and not the discussion. This part of the analysis is incomplete and the conclusions are not fully supported by the data. Is the overlap significantly higher than expected by chance? How many of the 10 mRNAs have different half-lifes following lincRNA-CCND1 KD and overexpression? Without these data the claim that this lincRNA can regulate other transcripts in the same way is not supported.

In order to further analyze the data of mRNA-seq and RIP-seq, we examined half-lives of 75 genes in the overlap (CNBP enriched 4-fold above the control levels) in HCT116 cells expressing control shRNA or LAST shRNA-1. In addition to CCND1 mRNA, at least three additional mRNAs were identified to be regulated by LAST. Their half-lives are indeed changed following lincRNA-CCND1 KD and overexpression (Figure 5). It suggests that CCND1 mRNA is not the only target regulated by LAST and CNBP. These data are now a part of the result in this revision (subsection “In addition to CCND1, LAST regulates the stabilization of some other mRNAs.”).

[Editors' note: further revisions were requested prior to acceptance, as described below.]

Major comments:1) It is not clear how the RIP-seq experiment was done – the authors refer to Yang et al., (2014), but it has RIP qRT-PCR and not RIP-seq. It is not clear if any digestion of the RNA following IP is performed. If there is no digestion – how can the authors conclude from it where CNBP is binding the CCND1 mRNA? If there is no digestion, then the peaks merely correspond to regions that are resistant to degradation, and it’s not surprising that they G/C-rich.

Reviewer is right, we performed RIP qRT-PCR as described in Yang et al., 2014, and RIP-seq was performed as described in Xiang et al.,(2014). The detailed method for RIP-seq has been added in the text (subsection “RIP-seq (RIP sequencing)”). In the RIP-seq experiment, cells were homogenized and then underwent 3 rounds of sonication on ice to remove the fragments that were not bound to CNBP. This information has been described in the method of RIP-seq.

2) The manuscript still needs very extensive language editing. Sentences like "In addition, c-Myc has been reported to repress transcription of CCND1 mediated by core elements of the CCND1 promoter but not MAX when expression of human Myc in BALB/c-3T3 cells, Rat6 cells and rat embryo fibroblasts." (from the Introduction) are not comprehendible. Other examples: "was particularly attracted our attention", "G1-relevant cyclins and CDKs genes", "To further confirm that LAST affect CCND1 mRNA stability is through CNBP", "It is therefore expected that knockdown of CNBP will reduced the association between LAST and CCND1 mRNA", "The exact reason for LAST or 5-'UTR involved in choosing the CNBP domain(s) for association is not clear at this moment", "Three mRNAs namely SOX9, NFE2L1 and PDF as the same as CCND1", "Six weeks after cells expressing shRNA injection or three weeks after cells expressing lncRNA injection, Due to the humane care for experimental animals,". These are just some of the examples. In present form, the article cannot be published in eLife without editing.

Sentences mentioned by the reviewer as some of the examples have been corrected (Introduction; subsection “Identification of LAST, a c-Myc-responsive long noncoding RNA that promotes cell proliferation”; subsection “LAST promotes G1/S transition and upregulates cyclin D1/CCND1”; subsection “LAST cooperates with CNBP to regulate CCND1mRNA stability”; subsection “Both LAST and CCND1 mRNA bind to CNBP through their G-rich motifs”; subsection “In addition to CCND1, LAST regulates the stability of other mRNAs.”; subsection “LAST promotes tumorigenesis). Furthermore, we had our whole manuscript to be edited by English language editing service.

3) The lack of any anti-correlation between CCND1 and LAST across tumor samples, which should be discussed in the manuscript. Its negative data, but important negative data, and so cannot be left out.

We have added these data in Supplementary file 4, and the description has been included in the Discussion section.

4) We cannot find the accession number for the mRNA-seq, microarray and RIP-seq results in the manuscript. Was the raw data deposited?

We have uploaded the mRNA-seq, microarray and RIP-seq raw data to Gene Expression Omnibus (GEO). The accession numbers are provided in the text (subsections “Identification of LAST, a c-Myc-responsive long noncoding RNA that promotes cell proliferation”, “RIP-seq (RIP sequencing)” and “mRNA-seq (mRNA sequencing)”).